# Exploring Representations and Interventions in Time Series Foundation Models

## Abstract

Time series foundation models (TSFMs) promise to be powerful tools for a wide range of applications. However, their internal representations and learned concepts are still not well understood. In this study, we investigate the structure and redundancy of representations across various TSFMs, examining the self-similarity of model layers within and across different model sizes. This analysis reveals block-like redundancy in the representations, which can be utilized for informed pruning to improve inference speed and efficiency. Additionally, we explore the concepts learned by these models—such as periodicity and trends—and how these can be manipulated through latent space steering to influence model behavior. Our experiments show that steering interventions can introduce new features, e.g., adding periodicity or trends to signals that initially lacked them. These findings underscore the value of representational analysis for optimizing models and demonstrate how conceptual steering offers new possibilities for more controlled and efficient time series analysis with TSFMs.

## 1 Introduction

Foundation models have taken significant strides in modeling both textual (Brown et al., 2020) and visual (Dosovitskiy et al., 2020) data, and have made complex language and image processing accessible to non-experts. These models are pre-trained on massive internet-scale datasets and can be used to solve multiple tasks across a variety of domains, with little to no adaptation. Recently, a growing body of work (Garza & Mergenthaler-Canseco, 2023; Goswami et al., 2024; Rasul et al., 2024; Das et al., 2024; Woo et al., 2024; Ansari et al., 2024) has extended the benefits of this paradigm to time series data, a modality prevalent in fields such as finance (Taylor, 2008), healthcare (Goswami et al., 2021), and climate science (Schneider & Dickinson, 1974).

Time series foundation models (TSFMs) have shown promising performance on multiple modeling tasks such as forecasting, classification, anomaly detection, and imputation, across a wide range of domains, and in settings with varying amounts of data and supervision. However, the underlying mechanisms and learned representations of TSFMs remain largely unexplored. Little is known about *the characteristics of learned representations, the kinds of concepts that these models are learning, and how can these concepts could be manipulated to influence model outputs.* A deeper understanding of the inner workings of TSFMs is key to enhancing their performance and trustworthiness. We address these knowledge gaps, by systematically *probing* these models and *intervening* in them.

We begin by analyzing TSFMs from two complementary perspectives: (1) *representational similarity* (Sec. 3.1), and *conceptual understanding* (Sec. 3.2). Our first set of experiments are aimed at answering the fundamental question: Are two TSFMs learning the same thing? We assess this using the similarity of representations learned by these models. The second set of experiments focus on identifying "which" human-interpretable concepts TSFMs learn, and "where" these concepts emerge. Our results in Sec. 4 show find that TSFMs learn redundant representations, and intuitive concepts such as base patterns (e.g., constant vs. sinusoidal waves), amplitudes, periodicity, and trends; often in specific layers and patches.

We leverage these insights to improve TSFMs and their trustworthiness in two ways: (1) In Sec. 3.1, we exploit their redundant representations to perform layer-wise pruning, which reduces model size, accelerates inference while preserving accuracy. (2) In the following Sec. 3.2, we steer model predictions along specific conceptual directions (e.g., introducing an upward trend in forecasts) using

synthetic time series as a way to imbue domain expertise into model predictions without explicit training. Our work is the first step towards understanding the inner workings of TSFMs and utilizing this knowledge to devise methods to improve their capabilities and controllability.

## 2 RELATED WORK

**Time Series Foundation Models (TSFMs).** TSFMs are versatile neural networks pre-trained on vast amounts of time series data, and have shown remarkable capabilities in producing accurate predictions even in zero-shot settings. Recently, a number of TSFMs have been proposed (Garza & Mergenthaler-Canseco, 2023; Liu et al., 2023; Das et al., 2024; Woo et al., 2024; Goswami et al., 2024; Ansari et al., 2024; Ekambaram et al., 2024; Talukder et al., 2024). While most TSFMs are based on variations of the Transformer architecture (Vaswani, 2017), they exhibit notable differences in tokenization strategies, pre-training datasets, and the specific tasks they are designed to address. While our methods are broadly applicable to Transformer-based TSFMs, we will primarily focus on analyzing `MOMENT` Goswami et al. (2024), `Chronos` Ansari et al. (2024), and `MOIRAI` Woo et al. (2024), three representative TSFMs that are fully open-source and offer distinct approaches to time series tokenization (patch vs. discrete), architecture (encoder-only vs. encoder-decoder), and pre-training objectives (imputation vs. forecasting).

**Analyzing Representations of Deep Learning Models.** Deep learning models often operate as black boxes, making their internal mechanisms and learned representations difficult to understand. One approach to gaining insights into these models is by comparing their intermediate representations. Similarity metrics can be used to determine the similarity or dissimilarity of representations at different stages of a model, revealing the hierarchy, homogeneity, and redundancy of learned features. Previous studies focused on quantifying the similarity of neural network representations Raghu et al. (2017); Kornblith et al. (2019). Raghu et al. (2021) demonstrated the use of these metrics for analyzing and comparing representations in vision transformers and CNNs, providing valuable insights into their functioning. Nguyen et al. (2021) investigated the impact of model size and training data ratios on similarity patterns by varying the depth and width of different models. They also explored model pruning based on similarity. Building on these these studies, our work presents the first comprehensive analysis of representations learned by TSFMs, offering useful insights into their internal workings.

**Identifying and Manipulating Learned Concepts in Pre-trained Models.** Understanding the internal representations learned by pre-trained models has been an active area of research, particularly in the context of LLMs and vision models. Previous studies have explored whether individual neurons or directions in a model's latent space correspond to specific features or concepts (Dalvi et al., 2019; Goh et al., 2021; Gurnee et al., 2023; Elhage et al., 2022). These investigations often focus on identifying linear representations, where features are encoded as linear combinations of neuron activations. Recent work has also employed various probing techniques to classify and interpret these internal representations, addressing aspects such as truthfulness and model robustness (Azaria & Mitchell, 2023; Zou et al., 2023; Burns et al., 2023; Marks & Tegmark, 2023). While many of these studies rely on meticulously curated datasets to probe language and vision models, we demonstrate that for time series models, synthetic data generated using simple mechanisms can effectively identify, localize, and probe the concepts learned by these models.

## 3 METHODS: PROBING AND INTERVENING IN TIME SERIES FOUNDATION MODELS

We study TSFMs using two complementary analysis and intervention methods. In Section 3.1, we examine learned representations through the lens of similarity, uncovering the redundancy inherent in TSFM representations. We leverage this redundancy to prune multiple layers of pre-trained models, thereby improving their efficiency without compromising accuracy. In Section 3.2, we identify the specific concepts learned by TSFMs, localizing them to specific hidden states. Furthermore, we explore the ability to steer model predictions along these conceptual directions, enabling us to influence model outputs in targeted ways.

## 3.1 PRUNING TIME SERIES FOUNDATION MODELS

### 3.1.1 ANALYZING REPRESENTATIONAL SIMILARITY FOR EFFECTIVE PRUNING

To gain a comprehensive understanding of the similarity between learned representations, we considered several metrics commonly employed in the literature. While our primary analysis relies on Centered Kernel Alignment (CKA) Kornblith et al. (2019), we also explored Cosine Similarity and Singular Vector Canonical Correlation Analysis (SVCCA) (Raghu et al., 2017). For brevity, we provide a brief overview of CKA below, while detailed descriptions of the remaining metrics can be found in Appendix B.

**Representational Similarity using Centered Kernel Alignment (CKA).** CKA measures the similarity of representations by comparing the centered kernel matrices. It has been shown to be effective in capturing similarities between layers of deep networks (Kornblith et al., 2019). The general form of CKA between two sets of representations $\mathbf{X}$ and $\mathbf{Y}$ is defined as:

$$\text{CKA}(\mathbf{X}, \mathbf{Y}) = \frac{\text{HSIC}(\mathbf{X}, \mathbf{Y})}{\sqrt{\text{HSIC}(\mathbf{X}, \mathbf{X}) \cdot \text{HSIC}(\mathbf{Y}, \mathbf{Y})}} \tag{1}$$

where HSIC denotes the Hilbert-Schmidt Independence Criterion (Gretton et al. (2005)).

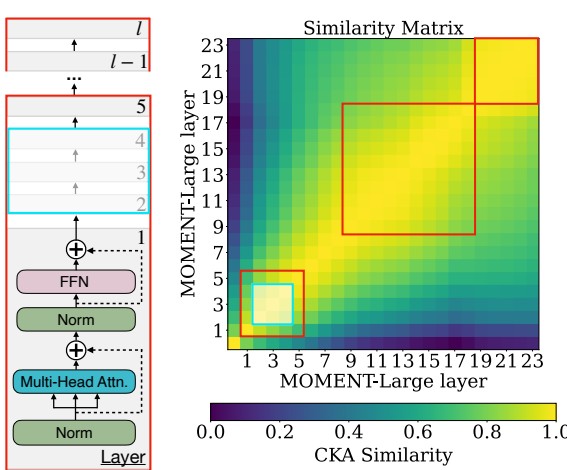

For computational efficiency, we utilized a linear kernel in our CKA calculations, resulting in the following simplified formula:

$$\text{CKA}_{\text{linear}}(\mathbf{X}, \mathbf{Y}) = \frac{\|\mathbf{X}^T \mathbf{Y}\|_F^2}{\|\mathbf{X}^T \mathbf{X}\|_F \cdot \|\mathbf{Y}^T \mathbf{Y}\|_F} \tag{2}$$

where $\| \cdot \|_F$ denotes the Frobenius norm. The denominator in Equation 2 ensures that the metric value falls within the range of 0 to 1, facilitating interpretability. A high CKA value indicates a strong alignment between the two sets of representations, suggesting that the layers are likely learning similar features or concepts.

Figure 1: For each identified block of layers exhibiting redundant representations (red), we remove the internal layers of the block by zeroing out their weights (blue). For example, if a block consists of five layers, we prune layers 2 through 4, retaining only the first and last layers to reduce representation redundancy while maintaining model integrity. More details on pruning can be found in App. C.

**Pruning TSFMs Based on Representational Similarity.** Large TSFMs typically learn redundant representations, which often manifest as block-like structures in heatmaps depicting pairwise similarity between layer activations (Figure 1). We can leverage this redundancy to downsize TSFMs, to improve their inference speed, without affecting their accuracy. We build on prior work (Nguyen et al., 2021) and propose a simple layer pruning strategy, which we call *Block-wise Pruning*, outlined in Algorithm 1. To preserve the structural integrity of each block, we retain the first and last layers of each block while zeroing out the weights of the intermediate layers. The skip connections within transformer blocks ensure that signals and gradients continue to flow through the rest of the network.

### 3.1.2 RESEARCH QUESTIONS AND EXPERIMENTAL SETUP

To gain a deeper understanding of TSFM representations, we investigate the following research questions: *(RQ1)* How similar are the representations learned by models of the same size but belonging to different families? *(RQ2)* How do these representations differ across models of varying sizes within the same family? *(RQ3)* How similar are the representations learned by corresponding layers of different TSFMs within the same family? To answer these questions, we use Centered

Kernel Alignment (CKA) to measure the similarity between representations at different layers of TSFMs, and visualize the results using heatmaps.

To demonstrate the effectiveness of our proposed pruning strategy, we explore two pruning configurations, one in which prune all redundant blocks, and the other where prune only a single block. We compare the performance of these pruned models to the original, unpruned TSFMs using standard task-specific accuracy metrics (Mean Squared Error and Mean Absolute Error) and efficiency metrics (inference time in milliseconds and theoretical model size in megabytes). We evaluate these models on widely used imputation (Zhou et al., 2021) and forecasting (Ansari et al., 2024) benchmarks in both zero-shot settings and after linear probing (Goswami et al., 2024).

**Algorithm 1:** Block-wise Pruning
___
**Require:** Trained model $\mathcal{M}$ with layers $\{l_1, l_2, \ldots, l_n\}$; Identified redundant blocks $\mathcal{B} = \{b_1, b_2, \ldots, b_k\}$
1: **for** each block $b_i$ in $\mathcal{B}$ **do**
2:    Let $b_i$ consist of layers $l_s$ to $l_e$ {Block edges at $l_s$ and $l_e$}
3:    **for** layer index $j = s + 1$ to $e - 1$ **do**
4:       Zero out the weights of layer $l_j$ in model $\mathcal{M}$
5:    **end for**
6: **end for**
7: **return** Pruned model $\mathcal{M}'$
___

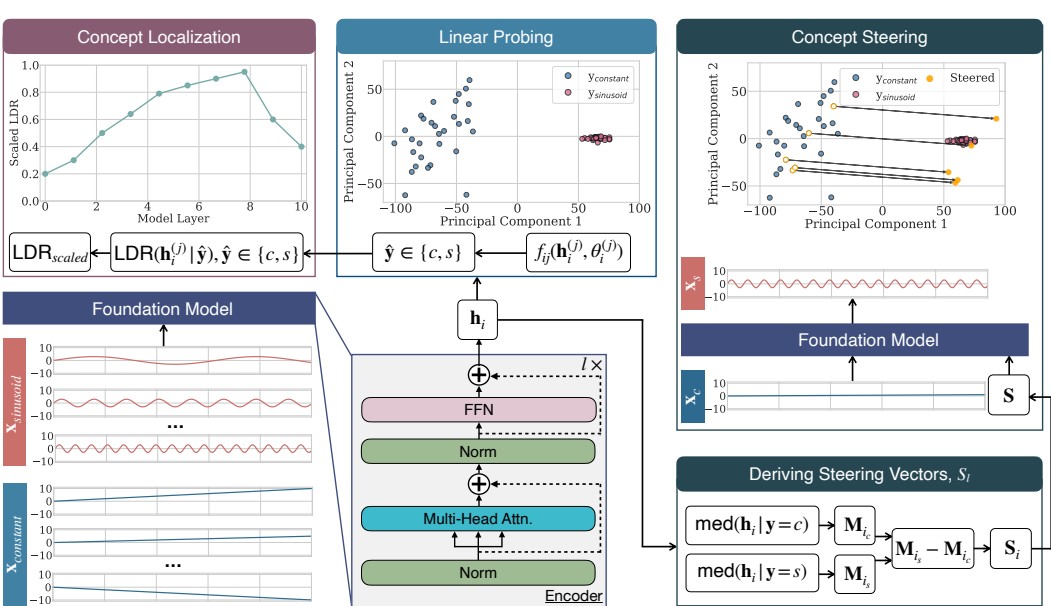

Figure 2: Overview of linear probing, concept localization, and steering. Linear probing involves training separate linear models for each layer or layer and patch to classify time series $\mathbf{x}$ into constant $c$ and sinusoid $s$ classes. Classifiers $f_{ij}(\mathbf{h}_i^{(j)}, \theta_i^{(j)})$ are trained on the hidden representation $\mathbf{h}_i^{(j)}$ at each $i$-th layer and $j$-th token to update the parameters $\theta_i^j$. Concept localization is achieved by computing Fisher's Linear Discriminant Ratio (LDR) between the classes at each layer and token using mean and variance statistics of $h_i^{(j)}$ for each predicted class, $\hat{y}$. The LDR output is scaled between 0 and 1 using min-max scaling to allow for consistent comparison across layers. Concept steering vector can be derived for each $i$-th layer by calculating the difference between the median activation matrices of the sinusoid and constant time series classes, $\mathbf{M}_{i_s} - \mathbf{M}_{i_c}$ and then stacked into a steering matrix $\mathbf{S}$ for the whole model. During model inference, the steering matrix can be used to steer model predictions towards desired concepts, or classes, by updating the embeddings as $\mathbf{h}_i \leftarrow \mathbf{h}_i + \lambda \mathbf{S}_i$, where $\lambda$ is a scalar that controls the strength of the intervention.

### 3.2 PROBING AND INTERVENING IN TIME SERIES FOUNDATION MODELS

#### 3.2.1 GENERATING SYNTHETIC DATA FOR CONCEPTUAL ANALYSIS OF TSFMs

To systematically explore the ability of TSFMs to understand intuitive time series concepts, we randomly generate a large number of synthetic univariate time series. Each randomly generated time series belong to one of two pattern classes: *constant* or *sinusoidal*. Constant patterns, represented by $y(t) = mt + b$, capture long-term non-periodic trends. Sinusoidal patterns, modeled as $y(t) = a\sin\left(\frac{2\pi t}{f}\right)$, represent periodic processes. By controlling the parameters $m$, $b$, $a$, and $f$, we can systematically generate time series with varying slope, intercept, amplitude, and periodicity, respectively. Despite their simplicity, these data generation mechanisms capture a wide range of real-world time series patterns. For a detailed description of the data generation process, please refer to Appendix A.

In this section, we build on the investigation approach outlined in (Marks & Tegmark, 2023). We say that a feature is linearly represented in a foundation model $\mathcal{M}$ if it is represented as a *direction* in its latent space. As a concrete example, consider that we want to identify whether $\mathcal{M}$ can distinguish between constant and sinusoidal patterns. If this feature is linearly represented in $\mathcal{M}$, we also want to identify which layer $l$ in $\mathcal{M}$ learns this concept in the most discriminant way.

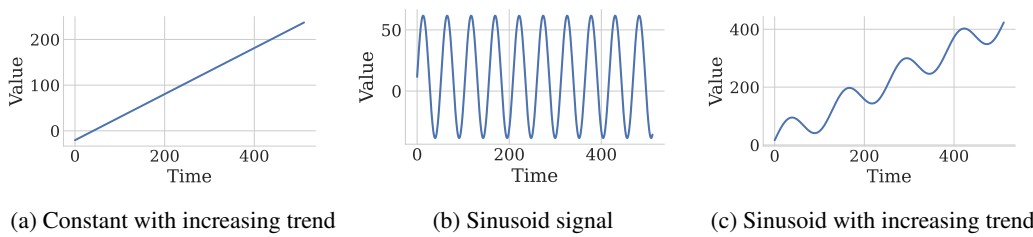

(a) Constant with increasing trend  (b) Sinusoid signal  (c) Sinusoid with increasing trend

Figure 3: Examples of synthetic data generated for experiments include constant signals with varying trend (a), sinusoidal signals with varying frequency (b), and compositions of constant signals with varying trends and sinusoidal signals, resulting in sinusoidal signals with varying trends (c). Synthetic data is used in linear probing, concept localization, and concept steering experiments.

#### 3.2.2 IDENTIFYING, LOCALIZING AND STEERING TIME SERIES CONCEPTS

**Identifying Linearly Represented Features.** To determine whether the feature (sinusoidal vs. constant time series) is linearly represented, we first generate a synthetic time series dataset by varying this feature. This dataset comprises of multiple sinusoids and constant time series randomly sampled using our data generating function. Using this dataset, we extract intermediate representations of each time series from the residual stream of each layer. Let $\mathbf{h}_i^{(j)} \in \mathbb{R}^{n \times D}$ denote the hidden representation of a time series $\mathbf{x}$ at $i$-th layer and $j$-th token of $\mathcal{M}$, where $D$ is the dimensionality of the hidden layer. Linear probing involves training separate linear models for each layer and token to classify time series $\mathbf{x}$ as a constant or sinusoid pattern. Classifiers $f_{ij}(\mathbf{h}_i^{(j)}, \theta_i^{(j)})$ are trained on the hidden representation $\mathbf{h}_i^{(j)}$ at each $i$-th layer and each $j$-th token to update the parameters $\theta_i^j$. Additionally, we perform probing on representations averaged along the token dimension for each $i$-th layer. The linear probes are trained to optimize the Fisher Criterion, a function that aims to maximize the distance between class means while minimizing within-class variance:

$$\mathcal{L}_{\text{Fisher}}(c, s) = -\frac{(\mu_s - \mu_c)^2}{\sigma_s^2 + \sigma_c^2}. \tag{3}$$

Here, $\mu_s$ and $\mu_c$ correspond to the mean embedding values, computed using all time series of a given class. Similarly, $\sigma_s^2$ and $\sigma_c^2$ correspond to the variance computed across the $n$ dimension for each class.

**Localizing Linearly Represented Features.** To localize which layers and tokens learn a specific concept, we compute the Fisher's Linear Discriminant Ratio (LDR) between the classes using the

mean and variance statistics of $\mathbf{h}_i^{(j)}$ for each predicted class $\hat{\mathbf{y}}$, which is determined using the classifier $f_{ij}$ during linear probing. The goal of LDR is to maximize the separation between the classes by comparing the variance $\sigma^2$ within each class to the difference between the class means, $\mu$. A larger ratio indicates a clearer separation between the two classes, which can aid in concept localization by identifying where the classes are well-separated in the feature space. When applied in the context of neural network activations, LDR helps highlight which layers or features are most discriminative

$$\text{LDR}(\mathbf{h}_i^{(j)}|\hat{\mathbf{y}}) = \frac{(\mu_{\mathbf{h}_i^{(j)}|\hat{\mathbf{y}}=s} - \mu_{\mathbf{h}_i^{(j)}|\hat{\mathbf{y}}=c})^2}{\sigma^2_{\mathbf{h}_i^{(j)}|\hat{\mathbf{y}}=s} + \sigma^2_{\mathbf{h}_i^{(j)}|\hat{\mathbf{y}}=c}} \tag{4}$$

$$= \frac{(\mu_s - \mu_c)^2}{\sigma_s^2 + \sigma_c^2}. \tag{5}$$

Here, $\mu_s$ and $\mu_c$ correspond to the mean computed across the $n$ dimension for each class. Similarly, $\sigma_s^2$ and $\sigma_c^2$ correspond to the variance computed across the sample dimension $n$ for each class. Let $\mathbf{V} = [v_{i,j}] \in \mathbb{R}^{L \times N}$ be the matrix of LDR values, where $v_{i,j}$ represents the LDR value for the $i$-th layer and $j$-th token, with $l$ layers and $N$ tokens. The LDR output is scaled between 0 and 1 using min-max scaling to allow for consistent comparison across layers. By visualizing the scaled LDA values as shown in Figure 2, one can identify which layers and tokens exhibit the highest degree of separation between classes, offering insights into the network's internal representations for concept intervention techniques.

**Deriving Steering Matrices for Model Steering.** Once we have identified that a feature is linearly represented in the latent space of the $\mathcal{M}$, we can use steering interventions to manipulate the latent space and generate time series that reflect intended concepts. For instance, to introduce periodicity to a constant time series, we can utilize a steering matrix $\mathbf{S}$, as illustrated in Figure 2. By strategically intervening in $\mathcal{M}$ using this steering matrix, we can bias its outputs towards predicting periodic time series. To construct a steering matrix, we first derive steering vectors $\mathbf{S}_i \in \mathbb{R}^{N \times D}$, for each layer $i$. These vectors represent the change that activations in layer $i$ must undergo such that $\mathcal{M}$ produces periodic outputs. $\mathbf{S}_i$ is simply the difference between the *median* activation matrix of the constant time series $\mathbf{M}_{i_c}$, from that of sinusoids $\mathbf{M}_{i_s}$. We stack these vectors for all layers to derive the steering matrix. This matrix allows us to simultaneously intervene across multiple tokens and layers during inference, which we found to be more effective than single-token interventions. During inference, to steer model predictions, at each layer $i$, we update its hidden representation as follows: $\mathbf{h}_i \leftarrow \mathbf{h}_i + \lambda \mathbf{S}_i$, where $\lambda \in \mathbb{R}$ is a scalar that controls the strength of the intervention.

### 3.2.3 RESEARCH QUESTIONS AND EXPERIMENTAL SETUP

Through our experiments, we aim to answer the following research questions: *(RQ4)* Do TSFMs represent concepts associated with specific data-generating functions distinctly in the latent space? *(RQ5)* Are these learned concepts localized to specific layers and tokens within TSFMs? *(RQ6)* Can we leverage these learned concepts to bias model predictions towards intended outcomes? For example, can we add periodicity or an upward trend to a constant time series? *(RQ7)* Is it possible to combine multiple steering interventions to manipulate model predictions towards complex compositions of various concepts? For instance, can we steer a model to add both trend and periodicity to a constant signal?

To address these research questions, we will leverage the techniques outlined in previous sections. We also explore two alternative modalities of intervention: (1) deriving steering vectors using the mean of hidden activations rather than the median, and (2) steering a single token versus all tokens throughout the model.

While our methods are broadly applicable to a wide range of transformer-based foundation models, we focus on two prominent TSFM families for brevity: MOMENT[1] (Goswami et al., 2024) and Chronos[2] (Ansari et al., 2024). Both these models are fully open-source, come in different sizes, yet have fundamentally different design choices. For example, Chronos is forecasting on encoder-decoder transformer model which takes discretized time series as input, whereas MOMENT is a multi-task, encoder-only model which takes continuous time series patches as input. Since only the Large

---

[1]https://github.com/moment-timeseries-foundation-model/moment
[2]https://github.com/amazon-science/chronos-forecasting

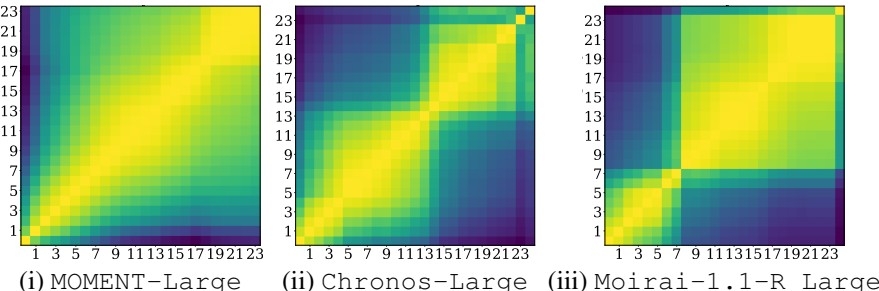

(i) MOMENT-Large    (ii) Chronos-Large    (iii) Moirai-1.1-R Large

Figure 5: Pairwise similarity of layer measured using CKA. Lighter shades indicate higher similarity (dark blue → low similarity, yellow → high similarity). While MOMENT shows substantial redundancy in representations, Chronos shows a more pronounced structure of block patterns. Moirai exhibits the most distinct block patterns, suggesting clearly defined stages of representation learning. These plots offer insights into localized regions of representation utility and redundancy across the three models.

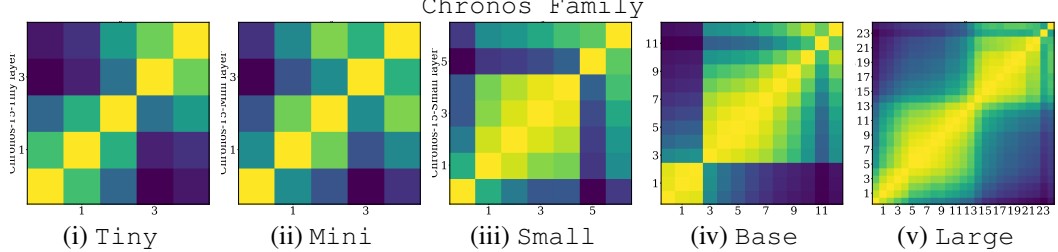

(i) Tiny    (ii) Mini    (iii) Small    (iv) Base    (v) Large

Figure 6: How does model size influence the patterns of learned representations? Smaller models exhibit more discrete and distinct block structures, while larger models (Base and Large) display increasingly intricate and clustered patterns, reflecting more nuanced and gradual transformations across layers. Notably, the emergence of blocks-like patterns in the Large model appears unpredictable from patterns observed in smaller models.

variant of MOMENT is publicly available at the time of writing this paper, we supplement our representation analysis results with Moirai (Woo et al., 2024)[3], another another popular TSFM which comes in different sizes. More information on model parameters can be found in Appendix E.

## 4 RESULTS

**Analyzing representations offers interesting insights.** Our analysis of model representations demonstrates that both model size and internal architecture considerably influence how representations are organized. Fig. 8) shows heatmaps which reveal that larger models, such as MOMENT-Large, Chronos-Large, and Moirai-1.1-R Large, have similar representations across specific groups of layers forming distinct and intricate block patterns, which may reflect unique stages of representation learning. More complex block patterns are observed with

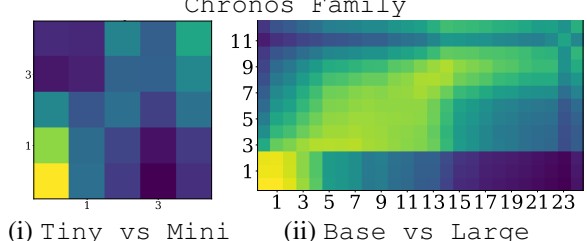

(i) Tiny vs Mini    (ii) Base vs Large

Figure 4: Similarity between representations learned by different layers in TSFMs of the same family but different sizes. Initial layers tend to learn similar representations, while the similarity gradually decreases in the later layers.

increasing model size, indicating that scaling may enhance the richness and organization of internal representations. However, it may also increase redundant knowledge storage through similar

---

[3]https://github.com/SalesforceAIResearch/uni2ts

representations across layers, as suggested by high CKA similarity measured in block patterns. Interestingly, within model families (e.g., `Chronos` and `Moirai`), scaling does not always result in predictable heatmap changes. Larger models, like `Chronos-Large` and `Moirai-Large`, demonstrate more refined and complex transformations of representations that are not easily extrapolated from their smaller versions as shown in Fig. 15). Moreover, cross-model similarity analysis results in Fig. 4 reveal that while early layers tend to have high similarity across models of different sizes, the similarity measures among later layers diverge more notably, particularly in larger models. This divergence is especially evident in the `Chronos` family, where early representations are more consistent across models, but later layers become increasingly specialized as model depth increases as shown in Fig. 6.

**Block-wise pruning can improve model throughput, without compromising accuracy.** We observed consistent improvements in memory efficiency and inference speed over their unpruned counterparts. For example, pruning only Block 3 for `MOMENT-Large`, resulted in a 11% decrease in estimated model size with a 5% speed up in inference. Furthermore, this pruned model had lower zero-shot imputation MAE for 5 of 9 datasets (ETTh2, ETTm1, ETTm2, Exchange, and Weather) as shown in Tab. 2. `Chronos-Large` results for zero-shot experiments are reported in Tab. 3. Detailed results on memory usage and speed improvements can be found in Tab. 6. While pruning consistently improved memory efficiency and inference speed compared to unpruned counterparts, performance varied across pruning methods and datasets, with some methods exhibiting considerable degradation. In addition to zero-shot experiments, we conducted experiments where models were fine-tuned post-pruning. For this, we applied `MOMENT-Large` for forecasting to compare a vanilla (unpruned) model to one with all block redundancies pruned, evaluating the impact of the most aggressive pruning approach. Fine-

| Dataset | Pruning | Forecasting Horizon | | | |
|---|---|---|---|---|---|
| | | 96 | 192 | 336 | 720 |
| Exchange | Vanilla | **0.109** | **0.215** | 0.417 | **1.003** |
| | All Pruned | 0.113 | 0.218 | **0.394** | 1.066 |
| ETTh1 | Vanilla | **0.385** | **0.411** | **0.423** | **0.443** |
| | All Pruned | 0.388 | 0.414 | 0.424 | 0.460 |
| ETTh2 | Vanilla | **0.287** | **0.350** | **0.370** | **0.404** |
| | All Pruned | 0.296 | 0.356 | 0.382 | **0.404** |
| ETTm1 | Vanilla | 0.290 | 0.330 | **0.352** | **0.409** |
| | All Pruned | **0.29** | **0.326** | 0.354 | 0.414 |
| ETTm2 | Vanilla | **0.171** | **0.231** | **0.287** | 0.372 |
| | All Pruned | 0.173 | 0.236 | 0.294 | **0.372** |
| ILI | Vanilla | 3.260 | 3.516 | 3.828 | 3.989 |
| | All Pruned | **2.981** | **3.209** | **3.479** | **3.602** |
| Weather | Vanilla | 0.153 | **0.197** | **0.246** | **0.316** |
| | All Pruned | **0.152** | 0.198 | 0.247 | 0.317 |

Figure 7: We fine-tune the vanilla and pruned variants on `MOMENT` on widely used long-horizon forecasting datasets (Zhou et al., 2021) and measure MSE. We found that the pruned model performed on par with the original model, underscoring the potential of our block-wise pruning approach. In this particular pruning setup, we reduced the memory consumption by more than half compared to the vanilla model and improved inference time per sample by ≈1 ms.

tuning results in Table 7 show that, notably, the pruned model performed nearly as well as the original, underscoring the potential of our block-wise pruning approach to maintain performance while reducing model complexity. Complete finetuning results are provided in Table 5 in Appendix F.

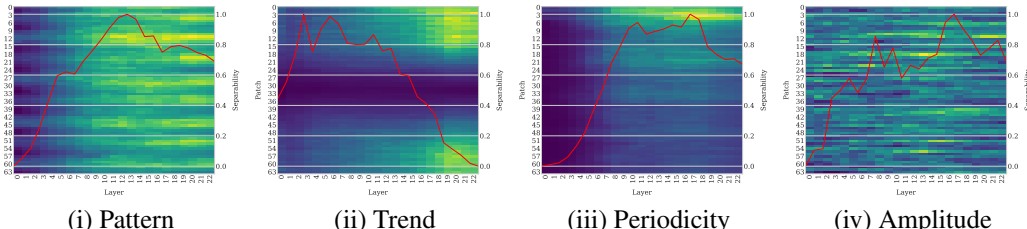

(i) Pattern       (ii) Trend       (iii) Periodicity       (iv) Amplitude

Figure 8: These heatmaps visualize the linear separability of various concepts at the patch level. Linear separability refers to the Linear Discriminant Ratio (LDR) computed from model embedding statistics for each predicted class: constant versus sinusoidal patterns (i), increasing versus decreasing trends (ii), high versus low periodicity (iii), and high versus low sinusoidal amplitude (iv). Lighter shades indicate higher separability (dark blue → low LDR, yellow → high LDR). *These heatmaps show that certain concepts represented by* `MOMENT-Large` *are linearly separable and that this separability is not consistent but rather emerges at specific layers in the model.*

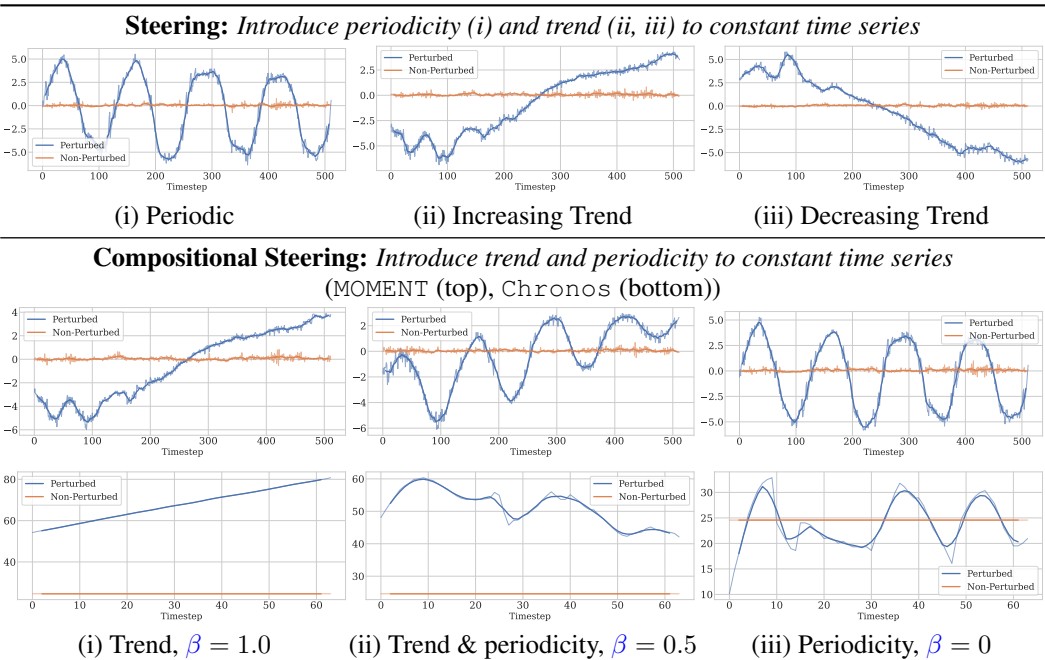

Figure 9: Visualization of the MOMENT's reconstruction and the Chronos's forecasting predictions (bottom), with concept steering applied in the latent space (blue) and the baseline without concept steering (orange). *Concept steering effectively transforms concepts in the latent space, resulting in model predictions that align with the intended concepts introduced.* Steering results are illustrated for the following experiments: steering a constant signal input to produce (i) a sinusoidal output, (ii) a constant signal with an increasing(slope $> 0$), and (iii) decreasing trend (slope $< 0$). For compositional steering experiments, $\alpha$ controls the strength of sinusoidal and increasing trend concepts. *When $\beta = 0.5$, models are steered towards a combination of increasing trend and sinusoidal pattern. $\beta = 0$ results in steering towards a constant signal with an increasing trend, whereas $\beta = 1$ only introduces sinusoidal patterns (iii).* Detailed results are available in the Appendix F.

**TSFMs learn Intuitive Linear Concepts.** Our concept localization results in Fig. 8 show that certain concepts represented by MOMENT-Large are linearly separable and that this separability is not consistent but rather emerges at specific layers in the model. We also found intuitive differences in the locations where these concepts are learned. We observed that certain concepts, such as distinguishing between constant and sinusoidal patterns, require careful examination of the entire time series. In contrast, differentiating between increasing and decreasing trends can be achieved by focusing on the initial and final patches. However, we did not identify specific locations where models learn to distinguish between time series of different amplitudes. This may be attributed to the normalization of input time series, a common practice in many TSFMs, including MOMENT.

**We can effectively steer TSFM predictions.** Our concept steering interventions effectively transform the latent space of TSFMs, resulting in model predictions that align with the intended concepts, as demonstrated in Fig. 9. We successfully introduced periodicity and trend concepts to constant time series and demonstrated the ability to combine multiple steering vectors to create more complex patterns. By combining steering vectors representing increasing trends and sinusoidal patterns, we were able to steer model predictions towards a combination of these features. To evaluate the effectiveness of steering in the latent space, we analyzed the impact of our interventions on hidden representation, by projecting in a two-dimensional space using Principal Component Analysis (PCA). We found that steering in the latent space is reflected in these lower-dimensional representations, as illustrated in Fig. 12. Notably, the PCA reduction often captured the concept direction as one of the principal components. This can be attributed to the careful design of our synthetic data generation process.

Interestingly, the method of obtaining the steering matrix, either by computing the mean or median across embedding concept classes, has no notable effect on the steered output as shown in Fig. 13. However, applying concept steering interventions across all tokens is necessary to achieve the intended steered concept output compared to applying concept steering interventions to a single token. Moreover, the $\lambda$ parameter can have considerable effect on steered output. For `Chronos`, steering required tuning the parameter $\lambda \approx 0.1$ for effective performance, whereas `MOMENT` maintained effective steering with $\lambda = 1$.

## 5 DISCUSSION

We explored two complementary approaches to probing and intervening in TSFMs. We gained valuable insights into their internal mechanisms and identified opportunities for improvement. For instance, our analysis revealed redundancy in learned representations. We leveraged this representational redundancy, inherent to large over-parameterized TSFMs, to devise a simple block-wise pruning strategy. This strategy effectively reduced the size and computational cost of these models without compromising performance, demonstrating the potential for distilling smaller, more efficient models from larger TSFMs.

We also explored ways to influence model predictions along conceptual directions using steering matrices. Concept steering has many practical applications, including the ability to correct prediction errors in pre-trained models that may arise from out-of-distribution inference, reflecting the effects of exogenous factors on model predictions or inducing prior knowledge not fully captured during training. Additionally, steering provides a method to reduce computational costs by minimizing the need for fine-tuning. It can also be used to introduce inductive biases from various domains into the model. This is particularly valuable for pre-trained models that may lack exposure to specific concepts due to limited or restricted training data. Such inductive biases can be useful in domains like healthcare, where, e.g., knowledge of the excitatory or inhibitory effects of treatments can guide pre-trained model predictions about whether a patient's vital signs should increase or decrease in response to specific interventions. Moreover, concept steering can be used for data generation, improving data augmentation techniques and creating more diverse datasets to imxprove TSFM performance across various tasks. Prior work has already shown that data augmentation techniques can improve TSFM model generalization by enhancing model robustness and exposing it to a wider variety of patterns (Ansari et al., 2024).

Finally, our findings also underscore the importance of synthetic data in studying and steering TSFMs. As opposed to meticulously curated datasets used to probe large language and vision models, we demonstrate that for time series models, synthetic data generated using simple mechanisms can effectively identify, localize, and probe the linear concepts learned by these models.

**Limitations and Future Work.** This paper provides insights into how a few basic patterns are linearly represented in time series foundation models. Future work must evaluate whether time series foundation models can learn more complex patterns present in real-world time series and whether steering matrices estimated using synthetic data can be used to steer predictions of out-of-distribution, real-world time series. While our methods are broadly applicable to different transformer-based foundation models, future research should explore other architectures such as state space models (Gu & Dao, 2023) and stacked multi-layer perceptrons (Ekambaram et al., 2024). Moreover, future studies should evaluate whether our findings hold for other time series foundation models and tasks such as anomaly detection, and classification. Beyond time series, we hope that our work inspires the use of synthetic data to steer large language and vision models as well.

## ETHICS STATEMENT

While our pruned TSFMs demonstrate promising performance, it is crucial to exercise caution when using them, especially in high-stakes applications such as healthcare. Before deploying these models for critical decision-making, we strongly recommend fine-tuning and evaluating them on task-specific, in-domain datasets using relevant metrics. The ability to steer model predictions offers numerous benefits but also raises concerns regarding potential biases. We urge users to exercise

caution when utilizing our proposed steering strategies and to carefully consider the potential implications of manipulating model outputs.

## REPRODUCIBILITY STATEMENT

To ensure reproduciblity, we have made our code anonymously accessible through `https://anonymous.4open.science/r/tsfm-interventions-3452/`. All time series foundation models used for our analysis are publicly available and open-source: `Chronos` (`https://github.com/amazon-science/chronos-forecasting`), `MOMENT` (`https://github.com/moment-timeseries-foundation-model/moment`), and `MOIRAI` (`https://github.com/SalesforceAIResearch/uni2ts`). All models were trained and evaluated on a computing cluster consisting of 128 AMD EPYC 7502 CPUs, 503 GB of RAM, and 8 NVIDIA RTX A6000 GPUs each with 49 GiB RAM. Synthetic datasets used in our study and the pruned time series foundation models will be released publicly upon paper acceptance.

## ACKNOWLEDGEMENTS

Omitted for double blind review.

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

## A  SYNTHETIC DATA GENERATION

Examples of synthetic data are provided in Fig. 10. Generated time series $\{x_t\}_{t=1}^{T}$ patterns include:

- **Constant Pattern.** The constant pattern captures long-term progression of the time series and is modeled as:
$$y(t) = mt + b,$$
where $\alpha$ is the slope and $\beta$ is the intercept. These parameters are sampled from uniform distributions:
$$m \sim U(\text{slope}_{\min}, \text{slope}_{\max}), \quad b \sim U(\text{intercept}_{\min}, \text{intercept}_{\max}).$$

- **Sinusoidal Pattern.** The sinusoidal pattern captures patterns of periodic variations and is modeled as:
$$y(t) = a \sin\left(\frac{2\pi t}{f}\right) + mt + b,$$
where $A$ is the amplitude and $P$ is the period, both sampled from uniform distributions:
$$a \sim U(\text{amplitude}_{\min}, \text{amplitude}_{\max}), \quad f \sim U(\text{period}_{\min}, \text{period}_{\max}).$$

Parameter Ranges:

- **Constant Case:**
$a \sim U(0,0), \quad f \sim U(0,0), \quad m \sim U(0,0), \quad b \sim U(-30,30).$
- **Increasing Slope Case:**
$a \sim U(0,0), \quad f \sim U(0,0), \quad m \sim U(0.5,1), \quad b \sim U(-30,30).$
- **Decreasing Slope Case:**
$a \sim U(0,0), \quad f \sim U(0,0), \quad m \sim U(-1,-0.5), \quad b \sim U(-30,30).$
- **Sine Constant Case (with seasonality parameters):**
$a \sim U(50,50), \quad f \sim U(128,128), \quad m \sim U(0,0), \quad b \sim U(-30,30).$
- **Sine Increasing Slope Case (with seasonality parameters):**
$a \sim U(50,50), \quad f \sim U(128,128), \quad m \sim U(0.5,1), \quad b \sim U(-30,30).$
- **Sine Decreasing Slope Case (with seasonality parameters):**
$a \sim U(50,50), \quad f \sim U(128,128), \quad m \sim U(-1,-0.5), \quad b \sim U(-30,30).$

## B  ADDITIONAL REPRESENTATION SIMILARITY METRICS

To comprehensively analyze the similarity of learned representations, we considered several metrics commonly used in the literature. While our primary analysis relies on Centered Kernel Alignment (CKA) Kornblith et al. (2019), we also explored two additional similarity metrics, namely Cosine Similarity and Singular Vector Canonical Correlation Analysis (SVCCA) (Raghu et al., 2017). Our findings consistently demonstrated similar patterns across all these metrics, underscoring the robustness of our results. We provide brief descriptions of Cosine Similarity and SVCCA below.

**Cosine Similarity:** Cosine similarity measures the cosine of the angle between two vectors, providing a simple yet effective way to assess similarity. In our case, we work with activation matrices for multiple samples and compute the average cosine similarity. Given two matrices $\mathbf{X}$ and $\mathbf{Y}$, representing the activations from two layers, the cosine similarity is computed as:

$$\text{cosine similarity}(\mathbf{X}, \mathbf{Y}) = \frac{1}{n} \sum_{i=1}^{n} \frac{\mathbf{x}_i \cdot \mathbf{y}_i}{\|\mathbf{x}_i\| \|\mathbf{y}_i\|} \tag{6}$$

where $\mathbf{x}_i$ and $\mathbf{y}_i$ are the $i$-th columns of matrices $\mathbf{X}$ and $\mathbf{Y}$, and $n$ is the number of samples.

**Singular Vector Canonical Correlation Analysis (SVCCA) (Raghu et al., 2017):** SVCCA is a method that compares the similarity of representations by aligning subspaces spanned by the top singular vectors. It effectively reduces the dimensionality and then compares the correlations of

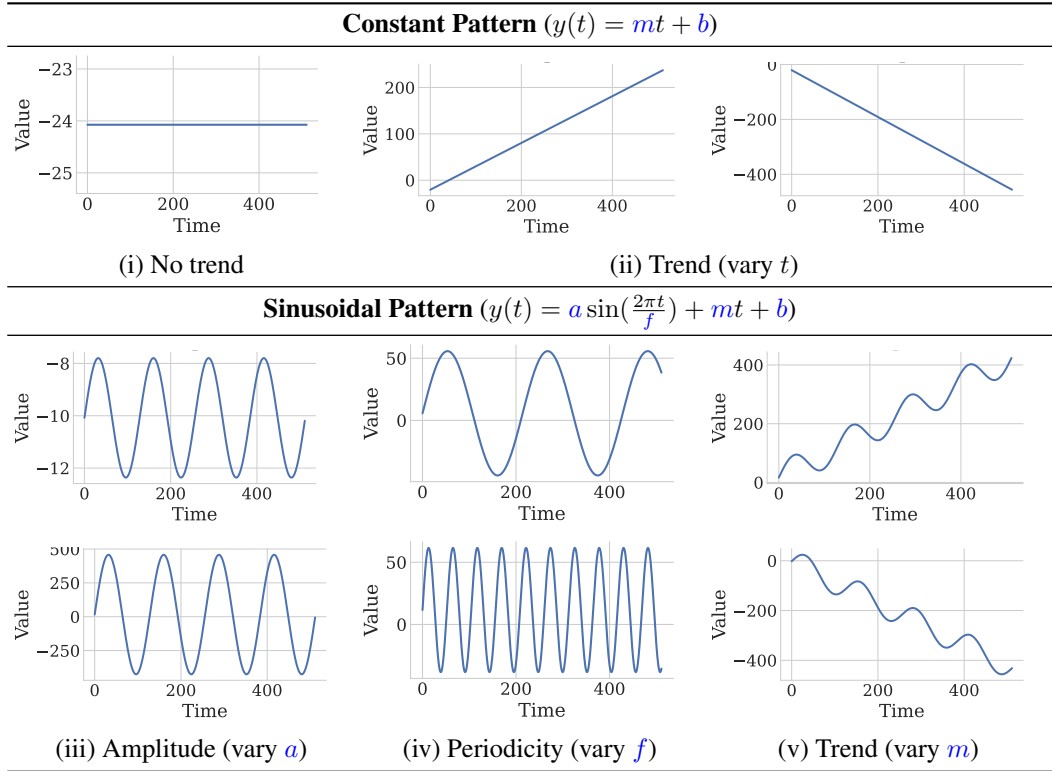

Figure 10: Samples of synthetic time series used in our experiments. We use two base patterns (constant and sinusoidal). To generate synthetic datasets, we vary the periodicity $f$, amplitude $a$, intercept $b$, and linear trend $m$.

the principal components. The SVCCA similarity between two activation matrices $\mathbf{X}$ and $\mathbf{Y}$ is computed as follows:

$$\text{SVCCA}(\mathbf{X}, \mathbf{Y}) = \text{CCA}(\mathbf{U}_k, \mathbf{V}_k) \tag{7}$$

where $\mathbf{U}_k$ and $\mathbf{V}_k$ are the top $k$ singular vectors obtained from the singular value decomposition (SVD) of $\mathbf{X}$ and $\mathbf{Y}$, respectively, and CCA denotes the canonical correlation analysis.

## C  FINDING AND PRUNING REDUNDANT BLOCKS IN TSFMS

The term "block" refers to groups of consecutive layers within a transformer that exhibit high representational similarity. Consistent with prior work Phang et al. (2021), we use "layer" to refer to individual transformer encoder or decoder blocks consistent with prior work, while "block" refers to a higher-level structure made up of multiple such layers that share similar representations Nguyen et al. (2021).

As an example, consider Figure 1 which illustrates the pairwise similarity between the 24 layers of `MOMENT-Large`. In this figure, lighter colors (yellow) represent higher representational similarity, as measured by Centered Kernel Alignment (CKA) Kornblith et al. (2019). The identified blocks are outlined with red bounding boxes. For example, Block 1 comprises layers 1–5, Block 2 comprises layers 9–18, and Block 3 comprises layers 19–23.

Our pruning algorithm is summarized in Algorithm 1. Our pruning method involves retaining the first and last layers of each block while zeroing out the weights of the intermediate layers. This approach preserves the structural integrity of the block while leveraging the skip connections within transformer blocks to ensure that signals and gradients continue to flow through the network. In the case of `MOMENT-Large`'s Block 3, composed of layers 19–23, this means that layers 19 and 23 are retained, while the weights of layers 20, 21, and 22 are zeroed out. We have added a dedicated section on pruning in Appendix C to further clarify this process.

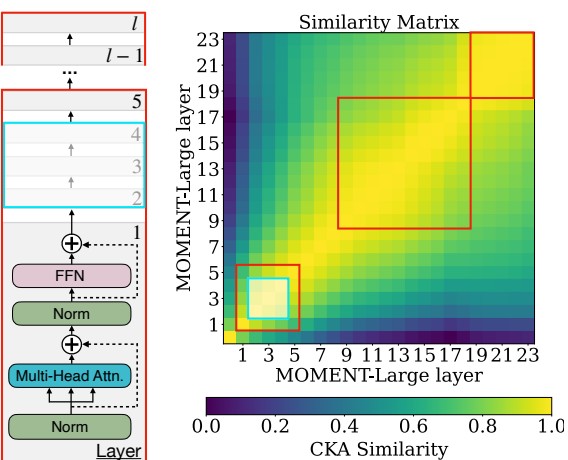

Figure 11: For each identified block of layers exhibiting redundant representations (red), we remove the internal layers of the block by zeroing out their weights (blue). For example, if a block consists of five layers, we prune layers 2 through 4, retaining only the first and last layers to reduce representation redundancy while maintaining model integrity.

---

**Algorithm 2:** Block Identification and Filtering

---

**Input:** CKA similarity matrix $S$, similarity threshold $\tau$, minimum block size $k$

1 blocks $\leftarrow \emptyset$;
2 current_block $\leftarrow \emptyset$;

3 **Phase 1: Initial block identification**;
4 **for** *each encoder block $l_i$* **do**
5      **if** $\forall l_j \in current\_block : CKA(l_i, l_j) \geq \tau$ **then**
6          current_block $\leftarrow$ current_block $\cup \{l_i\}$;
7      **end**
8      **else**
9          **if** $|current\_block| \geq k$ **then**
10              blocks $\leftarrow$ blocks $\cup \{current\_block\}$;
11          **end**
12          current_block $\leftarrow \{l_i\}$;
13      **end**
14 **end**

15 **Phase 2: Filter small blocks**;
16 filtered_blocks $\leftarrow \{b \in$ blocks $: |b| \geq k\}$;

17 **Phase 3: Verify block-wide self-similarity**;
18 final_blocks $\leftarrow \emptyset$;
19 **for** *each block $b \in$ filtered_blocks* **do**
20      start, end $\leftarrow$ indices of first and last layer in $b$;
21      submatrix $\leftarrow S[$start : end, start : end$]$;
22      min_similarity $\leftarrow \min($submatrix$)$;
23      **if** *min_similarity* $\geq \tau$ **then**
24          final_blocks $\leftarrow$ final_blocks $\cup \{b\}$;
25      **end**
26 **end**
27 **return** *final_blocks*

---

## C.1    A SIMPLE ALGORITHMIC APPROACH TO IDENTIFY REDUNDANT BLOCKS

We identify redundant blocks in a TSFM through visual inspection, which aligns with prior work Nguyen et al. (2021). Table 1 lists all the identified blocks in `MOMENT-Large` and `Chronos-Large`. In addition to visual inspection, redundant blocks can also be identified using algorithmic approaches.

Below we propose a simple algorithm to identify redundant blocks in TSFMs. First, we can define a block as:

$$\text{Block} = \{l_i, ..., l_j\} \text{ where } i < j \text{ and } l_k \text{ are adjacent transformer encoder blocks}$$

Then, we can systematically identify these blocks using an algorithm that:

1. Identifies initial candidate blocks by grouping adjacent layers with pairwise CKA similarity above a threshold
2. Filters out blocks that are too small to be considered meaningful
3. Verifies that each block exhibits high similarity across all its constituent layers by examining the complete submatrix of similarities

## C.2 IDENTIFIED BLOCKS IN MOMENT & CHRONOS

In this paper, identified redundant blocks visually. Below we specify, the redundant blocks for `MOMENT-Large` and `Chronos-Large`:

| Models | Block 1 | Block 2 | Block 3 | Block 4 |
|---|---|---|---|---|
| MOMENT-Large | $1-5$ | $9-18$ | $19-23$ | N/A |
| Chronos-Large | $1-4$ | $5-9$ | $10-13$ | $15-22$ |

Table 1: Visually identified blocks of redundant layers in `MOMENT-Large` and `Chronos-Large`.

## D ON STEERING AND THE PARAMETER $\lambda$

Below we provide some guidance on selecting good values of $\lambda$ and insights into its properties:

**Selection and Impact of the Steering Strength Parameter $\lambda$**

- **Optimal Range:** Based on our empirical experiments, we found that the steering strength parameter $\lambda$ is most effective for interventions when its value lies within the interval $[0.1, 2.0]$.
- **Lower Bound Considerations:** Values of $\lambda < 0.1$ often result in insufficient perturbation of the activation patterns, leading to suboptimal intervention effects that may not manifest visibly in the model's output.
- **Upper Bound Effects:** Setting $\lambda > 2.0$ induces excessive perturbations that push activations beyond their typical distribution bounds, potentially resulting in degenerate or semantically meaningless outputs. In the PCA/latent space visualizations, these cases simply appear as more distant points along the steering direction.

**Directional Properties**

- **Reversibility**: Multiplying the steering vector by -1 effectively reverses the direction of intervention, enabling bidirectional control (e.g., transforming concept $A \rightarrow B$ into $B \rightarrow A$).
- **Example application**: For a steering vector trained to increase signal magnitude, applying its negative counterpart ($-\lambda S$) produces controlled signal decrease, demonstrating the symmetric nature of the steering operation.

**Practical Guidelines**

- **Initial Calibration**: We recommend starting with $\lambda = 1.0$ and adjusting based on the observed intervention strength. In most cases value $\lambda = 1.0$ works well and does not need tuning.

- **Task and Model-Specific Tuning**: If $\lambda = 1.0$ does not yield satisfactory results, the optimal value requires tuning based on both the specific steering objective and target model, necessitating empirical calibration to achieve the desired intervention strength.
- **Monitoring**: When applying steering interventions, practitioners should monitor both the immediate output and latent space representations to ensure meaningful transformations while maintaining output coherence.

# E    MODEL DESCRIPTIONS AND SPECIFICATIONS

## E.1    MOMENT

The MOMENT model family is designed for general-purpose time-series analysis, utilizing an **encoder-only** Transformer architecture. It processes input time series by dividing them into fixed-length sub-sequences (patches) and encoding each patch into a D-dimensional space. The pre-training task involves reconstructing masked patches to learn robust representations that generalize across various tasks.

- **Architecture**: Encoder-only Transformer, with patch embeddings and masking for reconstruction.
- **Input Representation**: Time series split into fixed-length patches, embedded into D-dimensional vectors.
- **Model Variants**: MOMENT-Large

## E.2    CHRONOS

Chronos models utilize a **sequence-to-sequence (encoder-decoder)** Transformer architecture based on the T5 model family. Time series are scaled and quantized into tokens, which are processed by the encoder. The decoder autoregressively predicts future time steps, generating tokens that are mapped back into numerical values.

- **Architecture**: Encoder-decoder Transformer based on T5, with a reduced vocabulary size of 4096 tokens.
- **Input Representation**: Time series quantized into discrete tokens for sequence modeling.
- **Model Variants**: Chronos is available in the following configurations:
  - `chronos-t5-tiny`: 8M parameters
  - `chronos-t5-mini`: 20M parameters
  - `chronos-t5-small`: 46M parameters
  - `chronos-t5-base`: 200M parameters
  - `chronos-t5-large`: 710M parameters

## E.3    MOIRAI

Moirai is a time-series foundation model built with an **encoder-only** Transformer architecture designed for universal forecasting. The model handles varying temporal resolutions with multiple patch size projection layers and uses any-variate attention for multivariate time series. A mixture distribution is employed to model probabilistic forecasts.

- **Architecture**: Encoder-only Transformer with any-variate attention for multivariate time-series forecasting.
- **Input Representation**: Time series processed using multi-patch size projections to handle different frequencies.
- **Model Variants**: Moirai is available in three configurations:
  - `Moirai-small`: 14M parameters
  - `Moirai-base`: 91M parameters
  - `Moirai-large`: 311M parameters

# F  ADDITIONAL RESULTS

## F.1  STEERING

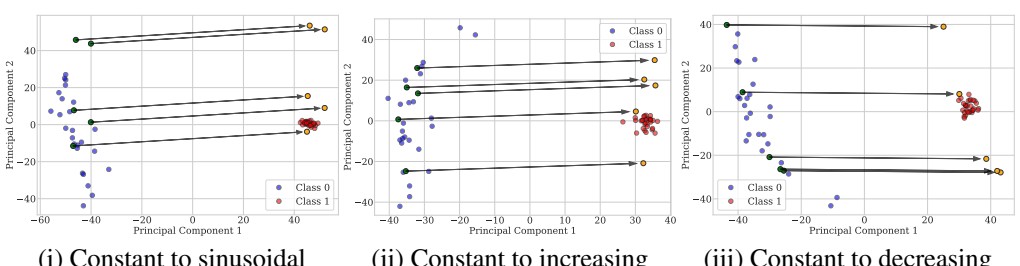

(i) Constant to sinusoidal      (ii) Constant to increasing      (iii) Constant to decreasing

Figure 12: Visualization of steering effects in latent space reduced using PCA analysis. Here, we considered steering constant to sinusoidal, constant to increasing, and constant to decreasing series. As shown, the steering behaves as expected in the embedding space, moving selected constant samples, into the neighborhood of sinusoidal/increasing/decreasing samples.

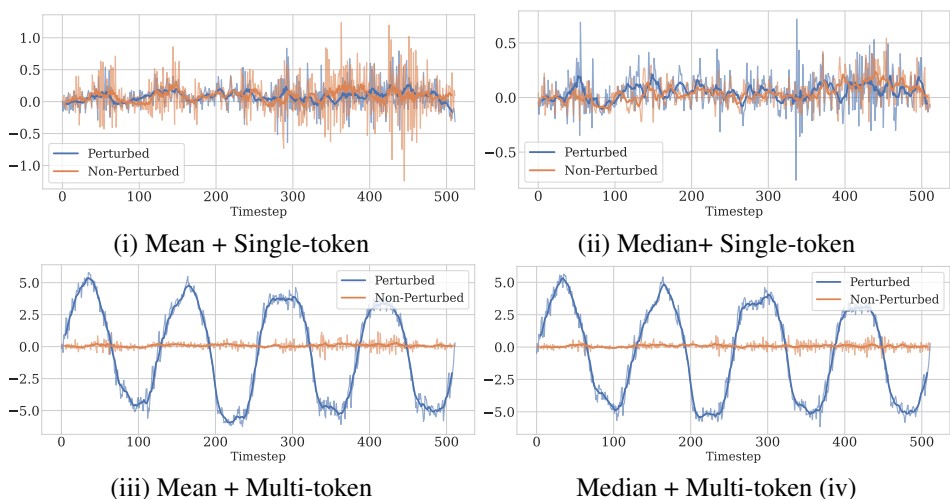

(i) Mean + Single-token      (ii) Median+ Single-token

(iii) Mean + Multi-token      Median + Multi-token (iv)

Figure 13: Visualization of different intervention and steering matrix derivation techniques. Applying concept steering interventions across all tokens is necessary to achieve the intended steered concept output (iii, iv) compared to applying concept steering interventions to a single token (i, iii). The method of obtaining the steering matrix, either by computing the mean or median across embedding concept classes, has no notable effect on the steered output.

**Application of Concept Steering on a Real-World Dataset**    We demonstrate the practical utility of concept steering on the ECG5000 dataset, which contains electrocardiogram readings classified as either normal or abnormal heart patterns. Using a MOMENT with an SVM classifier, we achieve strong baseline performance in distinguishing between the two classes.

For the steering experiment, we compute steering matrices using the median method to capture the concept difference between normal and abnormal patterns. Analysis in the activation space reveals that our steering approach successfully moves samples toward the target concept (Figure 17). This bidirectional movement is evident the PCA visualization of activation patterns.

To validate our approach quantitatively, we applied the steering transformations to 30 samples. While these samples were initially classified correctly with 100% accuracy as normal heartbeats, after steering, all of these samples swapped output classes, confirming successful concept transfer. These results suggest that concept steering can effectively capture and manipulate clinically relevant patterns in physiological data.

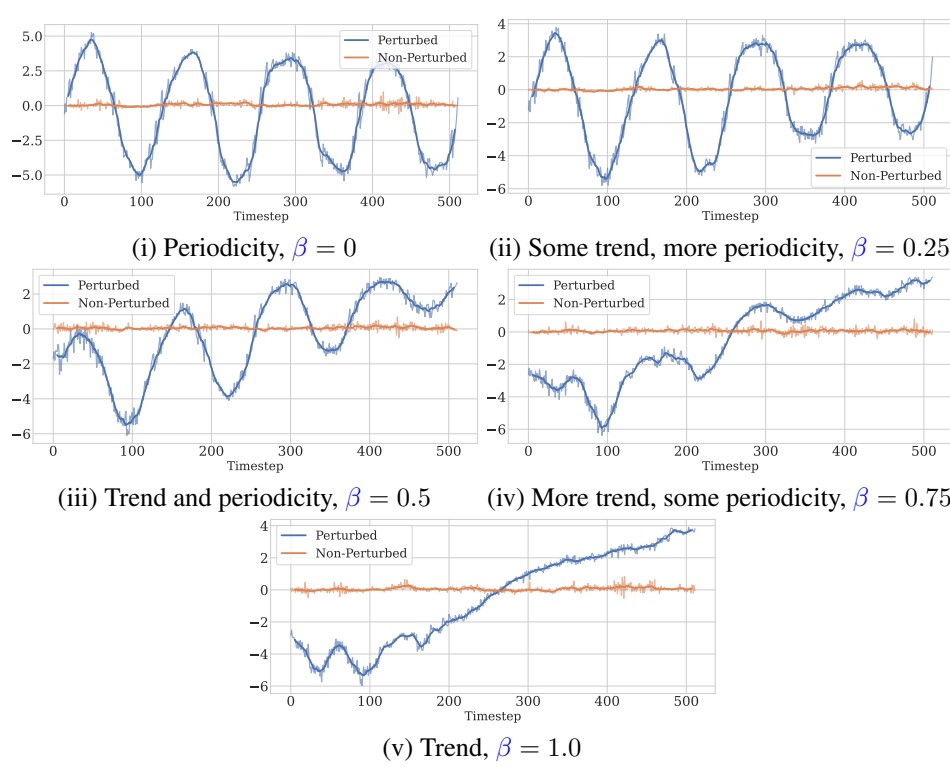

(i) Periodicity, $\beta = 0$  (ii) Some trend, more periodicity, $\beta = 0.25$

(iii) Trend and periodicity, $\beta = 0.5$  (iv) More trend, some periodicity, $\beta = 0.75$

(v) Trend, $\beta = 1.0$

Figure 14: Compositional Steering for Forecasting using the `MOMENT` model. The parameter $\beta$ influences compositional steering, where both sinusoidal and increasing trend concepts are observed in the steered output when $\beta$=0.5, showcasing how the steering technique can interpolate between different concept combinations in the latent space (ii). Steered output changes from sinusodial concepts to increasing trend concepts by varying $\beta \in \{0.0, 0.25, 0.5, 0.75, 1.0\}$.

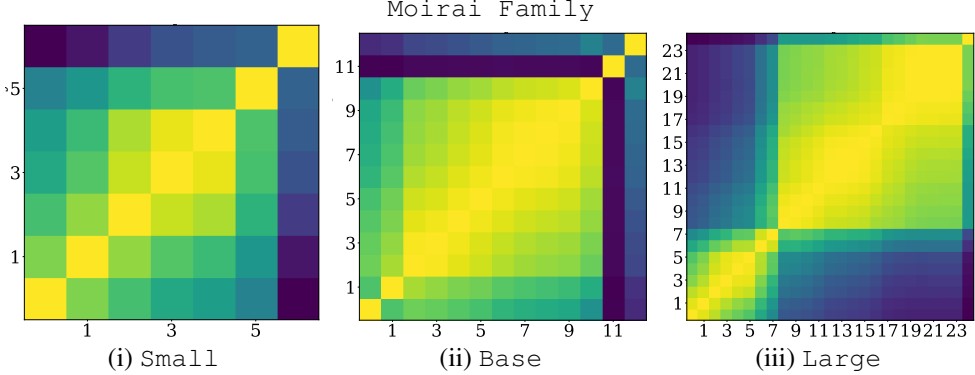

(i) `Small`  (ii) `Base`  (iii) `Large`

Figure 15: Self-similarity heatmaps comparing layer-to-layer CKA similarity matrices across different-sized models in the `Moirai` family. The Small and Base models exhibit similar patterns, with a distinct dissimilar layer at the end. The Base model has one additional dissimilar layer just before the final one. In contrast, the Large model presents a notably different structure, showing a two-block pattern where the layers are divided into two distinct groups, with the first block comprising roughly one-third of the model, and the second block covering the remaining two-thirds. This indicates a more pronounced hierarchical representation in the larger model.

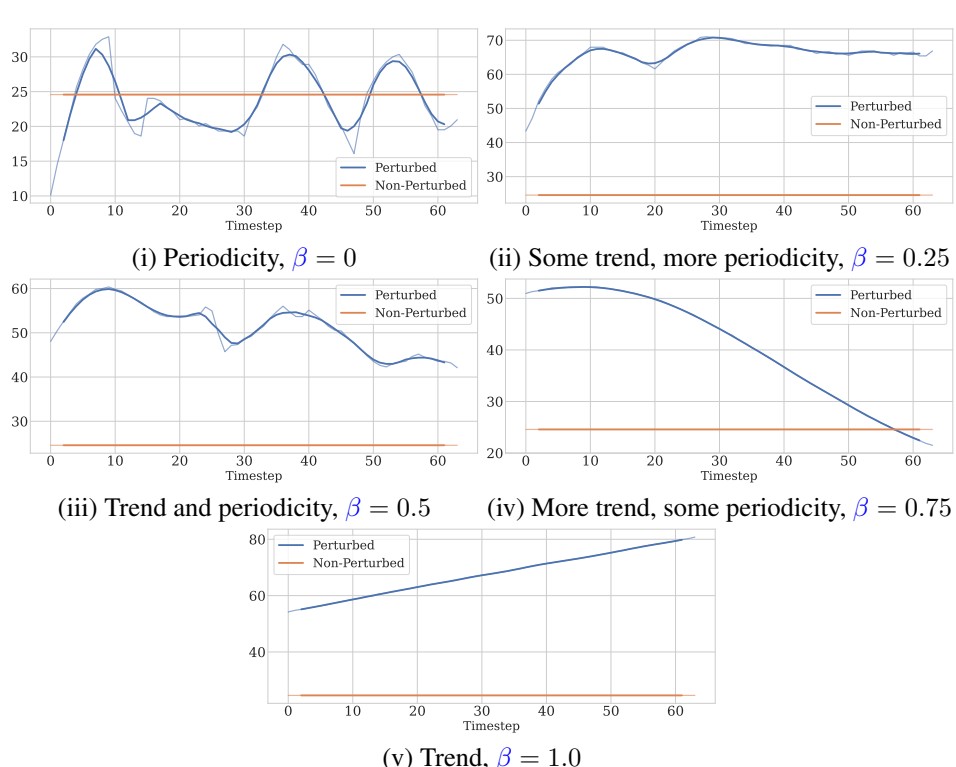

Figure 16: Compositional Steering for Forecasting using the `Chronos` model. The parameter $\beta$ influences compositional steering, where both sinusoidal and increasing trend concepts are observed in the steered output when $\beta$=0.5, showcasing how the steering technique can interpolate between different concept combinations in the latent space (ii). Steered output changes from sinusoidal concepts to increasing trend concepts by varying $\beta \in \{0.0, 0.25, 0.5, 0.75, 1.0\}$.

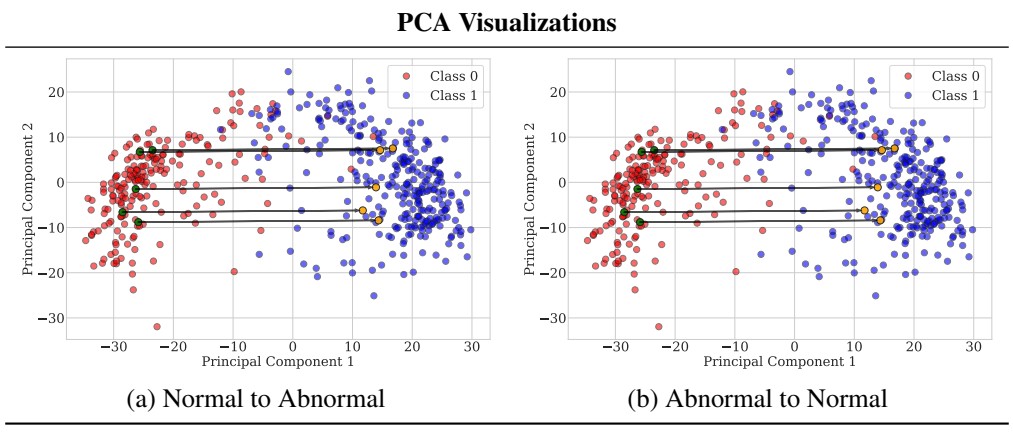

Figure 17: **Visualization of Concept Steering on ECG Data.** The figure shows PCA projections of the activation space, where arrows indicate the direction of steering. Blue and red points represent normal (class 0) and abnormal (class 1) samples respectively, with green points showing the original samples and orange points showing their steered versions.

## F.2 PRUNING

| Model name | Vanilla | | Block 1 | | Block 2 | | Block 3 | | All | |
|---|---|---|---|---|---|---|---|---|---|---|
| | MAE | MSE | MAE | MSE | MAE | MSE | MAE | MSE | MAE | MSE |
| ETTh1 | **0.395** | **0.371** | 0.448 | 0.449 | 0.521 | 0.614 | 0.420 | 0.424 | 0.548 | 0.673 |
| ETTh2 | **0.243** | **0.132** | 0.268 | 0.153 | 0.290 | 0.176 | **0.243** | 0.133 | 0.296 | 0.185 |
| ETTm1 | **0.287** | **0.204** | 0.321 | 0.233 | 0.358 | 0.307 | **0.287** | 0.221 | 0.345 | 0.284 |
| ETTm2 | 0.185 | 0.080 | 0.198 | 0.088 | 0.218 | 0.107 | **0.178** | **0.076** | 0.220 | 0.112 |
| Electricity | **0.372** | **0.250** | 0.446 | 0.342 | 0.716 | 0.757 | 0.428 | 0.327 | 0.727 | 0.819 |
| Exchange rate | 0.125 | 0.034 | 0.124 | 0.032 | 0.170 | 0.061 | **0.111** | **0.027** | 0.174 | 0.067 |
| Illness | **0.393** | **0.421** | 0.448 | 0.502 | 0.547 | 0.669 | 0.423 | 0.446 | 0.552 | 0.648 |
| Traffic | **0.492** | **0.790** | 0.552 | 0.906 | 0.861 | 1.562 | 0.606 | 0.940 | 0.878 | 1.633 |
| Weather | 0.129 | 0.079 | 0.134 | 0.082 | 0.170 | 0.107 | **0.117** | **0.074** | 0.176 | 0.120 |

Table 2: **Zero-shot imputation performance of `MOMENT`.** Results averaged across four different masking rates: {12.5%, 25%, 37.5%, 50%} and five runs with different masking seeds. This table presents the Model Performance Metrics (Mean Absolute Error and Mean Squared Error) on a subset of the Time Series Pile (Goswami et al., 2024; Zhou et al., 2021). The model names include: "Vanilla" `MOMENT-Large` without any pruning, "Block 1-3" for cases where only one block is pruned, and "All" for all three blocks being pruned. The best results per dataset are bolded. For full results refer to 4

| | Dataset | Vanilla | | Block 1 | | Block 2 | | Block 3 | | Block 4 | | All | |
|---|---|---|---|---|---|---|---|---|---|---|---|---|---|
| | | MASE | WQL | MASE | WQL | MASE | WQL | MASE | WQL | MASE | WQL | MASE | WQL |
| 0 | ETTh | 0.776 | **0.076** | 0.848 | 0.088 | **0.765** | 0.078 | 0.778 | 0.079 | 0.899 | 0.098 | 1.093 | 0.100 |
| 1 | ETTm | **0.716** | 0.068 | 0.820 | 0.076 | 0.864 | 0.084 | 0.721 | **0.063** | 0.724 | 0.074 | 0.978 | 0.084 |
| 2 | dominick | 0.820 | **0.331** | 0.829 | 0.335 | 0.824 | 0.334 | **0.809** | 0.332 | 0.835 | 0.337 | 0.886 | 0.359 |
| 3 | ercot | 0.627 | 0.020 | 0.694 | 0.022 | 0.697 | 0.027 | **0.624** | **0.018** | 0.909 | 0.024 | 1.281 | 0.042 |
| 4 | exchange_rate | 2.310 | **0.012** | 2.394 | 0.014 | 2.061 | 0.013 | 2.256 | 0.014 | 1.877 | 0.012 | **1.713** | 0.013 |
| 5 | m4_quarterly | 1.217 | 0.082 | 1.255 | 0.084 | **1.193** | **0.081** | 1.223 | 0.082 | 1.213 | 0.082 | 1.270 | 0.085 |
| 6 | m4_yearly | 3.559 | 0.133 | 3.537 | 0.132 | 3.495 | 0.129 | 3.507 | 0.131 | 3.502 | 0.129 | **3.359** | **0.123** |
| 7 | m5 | 0.943 | **0.586** | 0.942 | 0.586 | 0.954 | 0.591 | **0.939** | 0.587 | 0.941 | 0.622 | 0.956 | 0.644 |
| 8 | monash_australian_electricity | 1.427 | 0.076 | 1.417 | 0.077 | **1.372** | **0.073** | 1.445 | 0.083 | 1.638 | 0.084 | 2.593 | 0.143 |
| 9 | monash_car_parts | 0.903 | 1.041 | 0.883 | 1.046 | 0.867 | 0.998 | 0.892 | 1.027 | 0.833 | 0.960 | **0.816** | **0.956** |
| 10 | monash_cif_2016 | 0.989 | 0.012 | 1.058 | 0.019 | **0.944** | 0.013 | 0.998 | 0.018 | 1.096 | **0.011** | 1.275 | 0.017 |
| 11 | monash_covid_deaths | 43.251 | 0.058 | 42.444 | **0.052** | 44.001 | 0.078 | 42.357 | 0.060 | **41.915** | 0.053 | 45.544 | 0.062 |
| 12 | monash_fred_md | 0.517 | 0.021 | 0.520 | 0.019 | 0.507 | 0.024 | 0.553 | 0.029 | **0.504** | **0.016** | 0.600 | 0.017 |
| 13 | monash_hospital | 0.704 | 0.056 | 0.724 | 0.057 | **0.690** | **0.055** | 0.703 | 0.055 | 0.718 | 0.058 | 0.726 | 0.060 |
| 14 | monash_m1_monthly | 1.075 | 0.127 | 1.117 | 0.132 | **1.057** | **0.125** | 1.107 | 0.130 | 1.222 | 0.141 | 1.321 | 0.155 |
| 15 | monash_m1_quarterly | 1.728 | 0.106 | 1.743 | 0.102 | 1.659 | 0.105 | 1.727 | **0.092** | 1.748 | 0.105 | 1.753 | 0.098 |
| 16 | monash_m1_yearly | 4.336 | 0.183 | 4.275 | 0.174 | 4.113 | 0.174 | 4.375 | **0.163** | 4.400 | 0.182 | **4.100** | 0.170 |
| 17 | monash_m3_monthly | 0.855 | 0.096 | 0.887 | 0.100 | **0.845** | **0.094** | 0.864 | 0.096 | 0.909 | 0.102 | 0.968 | 0.108 |
| 18 | monash_m3_quarterly | 1.183 | 0.075 | 1.204 | 0.075 | **1.178** | **0.073** | 1.199 | 0.075 | 1.204 | 0.074 | 1.301 | 0.077 |
| 19 | monash_m3_yearly | 3.034 | 0.145 | 2.972 | 0.146 | 2.908 | 0.143 | 2.980 | 0.143 | 3.017 | 0.144 | **2.870** | **0.137** |
| 20 | monash_nn5_weekly | 0.936 | **0.090** | 0.961 | 0.092 | **0.924** | 0.090 | 0.939 | 0.090 | 0.941 | 0.090 | 0.985 | 0.095 |
| 21 | monash_tourism_monthly | 1.746 | 0.099 | 1.847 | 0.101 | **1.613** | **0.088** | 1.996 | 0.103 | 2.178 | 0.191 | 2.970 | 0.217 |
| 22 | monash_tourism_quarterly | 1.647 | 0.072 | 1.729 | 0.065 | **1.615** | **0.059** | 1.666 | 0.063 | 2.011 | 0.072 | 2.034 | 0.071 |
| 23 | monash_tourism_yearly | **3.565** | 0.180 | 3.683 | 0.185 | 3.574 | **0.173** | 3.724 | 0.188 | 3.596 | 0.174 | 3.568 | 0.175 |
| 24 | monash_traffic | **0.794** | 0.253 | 0.808 | **0.247** | 0.797 | 0.253 | 0.808 | 0.255 | 0.916 | 0.275 | 1.046 | 0.290 |
| 25 | monash_weather | **0.819** | **0.139** | 0.900 | 0.153 | 0.826 | 0.140 | 0.845 | 0.144 | 1.013 | 0.173 | 1.002 | 0.170 |
| 26 | nn5 | 0.574 | 0.157 | 0.584 | 0.160 | 0.599 | 0.165 | **0.571** | **0.154** | 0.792 | 0.219 | 0.905 | 0.255 |

Table 3: **Zero-shot forecasting performance of `Chronos-Large`.** This table presents zero-shot performance evaluated with Mean Absolute Scaled Error (MASE) and Weighted Quantile Loss (WQL). Results are presented for the model without any pruning (Vanilla), when individual blocks pruned Block i, and when all blocks are pruned (All).

| Dataset | Mask Ratio | All Mean MAE | All Mean MSE | All Std MAE | All Std MSE | Block 1 Mean MAE | Block 1 Mean MSE | Block 1 Std MAE | Block 1 Std MSE | Block 2 Mean MAE | Block 2 Mean MSE | Block 2 Std MAE | Block 2 Std MSE | Block 3 Mean MAE | Block 3 Mean MSE | Block 3 Std MAE | Block 3 Std MSE | Vanilla Mean MAE | Vanilla Mean MSE | Vanilla Std MAE | Vanilla Std MSE |
|---|---|---|---|---|---|---|---|---|---|---|---|---|---|---|---|---|---|---|---|---|---|
| ETTh1 | 0.125 | 0.541 | 0.662 | 0.016 | 0.069 | 0.448 | 0.446 | 0.023 | 0.084 | 0.516 | 0.608 | 0.016 | 0.072 | 0.426 | 0.425 | 0.038 | 0.119 | 0.398 | 0.370 | 0.034 | 0.091 |
|  | 0.25 | 0.548 | 0.667 | 0.015 | 0.049 | 0.445 | 0.441 | 0.019 | 0.056 | 0.520 | 0.605 | 0.021 | 0.073 | 0.418 | 0.420 | 0.026 | 0.073 | 0.393 | 0.365 | 0.026 | 0.061 |
|  | 0.375 | 0.550 | 0.674 | 0.015 | 0.035 | 0.447 | 0.446 | 0.011 | 0.033 | 0.522 | 0.612 | 0.016 | 0.045 | 0.418 | 0.422 | 0.019 | 0.056 | 0.394 | 0.368 | 0.014 | 0.033 |
|  | 0.5 | 0.551 | 0.687 | 0.011 | 0.036 | 0.452 | 0.463 | 0.012 | 0.037 | 0.525 | 0.630 | 0.011 | 0.041 | 0.420 | 0.429 | 0.011 | 0.035 | 0.397 | 0.380 | 0.010 | 0.027 |
|  | mean | 0.548 | 0.673 | 0.014 | 0.046 | 0.448 | 0.449 | 0.016 | 0.052 | 0.521 | 0.614 | 0.015 | 0.056 | 0.420 | 0.424 | 0.024 | 0.071 | 0.395 | 0.371 | 0.021 | 0.054 |
| ETTh2 | 0.125 | 0.290 | 0.179 | 0.005 | 0.016 | 0.269 | 0.155 | 0.008 | 0.016 | 0.292 | 0.177 | 0.010 | 0.020 | 0.240 | 0.130 | 0.010 | 0.015 | 0.243 | 0.131 | 0.007 | 0.015 |
|  | 0.25 | 0.298 | 0.188 | 0.008 | 0.015 | 0.271 | 0.159 | 0.011 | 0.019 | 0.294 | 0.182 | 0.010 | 0.019 | 0.245 | 0.136 | 0.010 | 0.015 | 0.247 | 0.136 | 0.012 | 0.017 |
|  | 0.375 | 0.296 | 0.186 | 0.005 | 0.009 | 0.266 | 0.151 | 0.011 | 0.015 | 0.286 | 0.172 | 0.008 | 0.013 | 0.245 | 0.135 | 0.007 | 0.010 | 0.242 | 0.130 | 0.011 | 0.014 |
|  | 0.5 | 0.298 | 0.187 | 0.005 | 0.010 | 0.266 | 0.150 | 0.008 | 0.011 | 0.288 | 0.174 | 0.006 | 0.011 | 0.242 | 0.132 | 0.006 | 0.008 | 0.242 | 0.129 | 0.007 | 0.010 |
|  | mean | 0.296 | 0.185 | 0.006 | 0.012 | 0.268 | 0.153 | 0.009 | 0.015 | 0.290 | 0.176 | 0.008 | 0.015 | 0.243 | 0.133 | 0.008 | 0.012 | 0.243 | 0.132 | 0.009 | 0.013 |
| ETTm1 | 0.125 | 0.344 | 0.281 | 0.013 | 0.039 | 0.322 | 0.236 | 0.012 | 0.023 | 0.361 | 0.313 | 0.016 | 0.035 | 0.284 | 0.216 | 0.008 | 0.026 | 0.288 | 0.208 | 0.011 | 0.025 |
|  | 0.25 | 0.343 | 0.280 | 0.010 | 0.026 | 0.319 | 0.230 | 0.012 | 0.026 | 0.357 | 0.304 | 0.012 | 0.029 | 0.284 | 0.216 | 0.008 | 0.024 | 0.284 | 0.199 | 0.013 | 0.028 |
|  | 0.375 | 0.345 | 0.283 | 0.005 | 0.014 | 0.319 | 0.230 | 0.005 | 0.013 | 0.356 | 0.302 | 0.006 | 0.016 | 0.288 | 0.222 | 0.004 | 0.011 | 0.286 | 0.199 | 0.006 | 0.016 |
|  | 0.5 | 0.347 | 0.292 | 0.004 | 0.010 | 0.322 | 0.238 | 0.004 | 0.009 | 0.357 | 0.309 | 0.002 | 0.006 | 0.291 | 0.229 | 0.002 | 0.010 | 0.288 | 0.207 | 0.003 | 0.008 |
|  | mean | 0.345 | 0.284 | 0.009 | 0.024 | 0.321 | 0.233 | 0.009 | 0.018 | 0.358 | 0.307 | 0.010 | 0.023 | 0.287 | 0.221 | 0.006 | 0.018 | 0.287 | 0.204 | 0.008 | 0.020 |
| ETTm2 | 0.125 | 0.222 | 0.113 | 0.005 | 0.008 | 0.200 | 0.089 | 0.004 | 0.002 | 0.221 | 0.110 | 0.005 | 0.004 | 0.179 | 0.078 | 0.005 | 0.004 | 0.188 | 0.082 | 0.005 | 0.004 |
|  | 0.25 | 0.220 | 0.112 | 0.006 | 0.005 | 0.197 | 0.086 | 0.005 | 0.004 | 0.216 | 0.105 | 0.006 | 0.005 | 0.177 | 0.074 | 0.004 | 0.003 | 0.184 | 0.078 | 0.004 | 0.003 |
|  | 0.375 | 0.219 | 0.112 | 0.003 | 0.003 | 0.198 | 0.088 | 0.003 | 0.002 | 0.216 | 0.106 | 0.004 | 0.003 | 0.178 | 0.075 | 0.002 | 0.002 | 0.184 | 0.080 | 0.003 | 0.002 |
|  | 0.5 | 0.219 | 0.111 | 0.001 | 0.002 | 0.198 | 0.088 | 0.003 | 0.003 | 0.218 | 0.107 | 0.004 | 0.004 | 0.178 | 0.075 | 0.002 | 0.001 | 0.185 | 0.080 | 0.003 | 0.003 |
|  | mean | 0.220 | 0.112 | 0.004 | 0.005 | 0.198 | 0.088 | 0.004 | 0.003 | 0.218 | 0.107 | 0.005 | 0.004 | 0.178 | 0.076 | 0.003 | 0.003 | 0.185 | 0.080 | 0.004 | 0.003 |
| electricity | 0.125 | 0.736 | 0.838 | 0.019 | 0.037 | 0.449 | 0.345 | 0.009 | 0.011 | 0.722 | 0.769 | 0.013 | 0.025 | 0.425 | 0.321 | 0.010 | 0.015 | 0.375 | 0.253 | 0.012 | 0.014 |
|  | 0.25 | 0.725 | 0.816 | 0.011 | 0.021 | 0.443 | 0.338 | 0.006 | 0.007 | 0.715 | 0.755 | 0.009 | 0.016 | 0.428 | 0.325 | 0.004 | 0.007 | 0.371 | 0.249 | 0.006 | 0.007 |
|  | 0.375 | 0.722 | 0.809 | 0.007 | 0.012 | 0.444 | 0.341 | 0.002 | 0.003 | 0.712 | 0.750 | 0.005 | 0.008 | 0.430 | 0.331 | 0.003 | 0.003 | 0.369 | 0.248 | 0.004 | 0.004 |
|  | 0.5 | 0.725 | 0.814 | 0.002 | 0.002 | 0.447 | 0.345 | 0.004 | 0.004 | 0.714 | 0.754 | 0.002 | 0.004 | 0.431 | 0.332 | 0.003 | 0.003 | 0.371 | 0.249 | 0.002 | 0.002 |
|  | mean | 0.727 | 0.819 | 0.012 | 0.023 | 0.446 | 0.342 | 0.006 | 0.007 | 0.716 | 0.757 | 0.009 | 0.016 | 0.428 | 0.327 | 0.006 | 0.009 | 0.372 | 0.250 | 0.007 | 0.008 |
| exchange rate | 0.125 | 0.164 | 0.059 | 0.010 | 0.014 | 0.129 | 0.035 | 0.007 | 0.006 | 0.178 | 0.065 | 0.010 | 0.008 | 0.109 | 0.027 | 0.010 | 0.002 | 0.131 | 0.037 | 0.010 | 0.006 |
|  | 0.25 | 0.172 | 0.064 | 0.010 | 0.012 | 0.122 | 0.031 | 0.006 | 0.003 | 0.167 | 0.059 | 0.001 | 0.002 | 0.108 | 0.025 | 0.007 | 0.001 | 0.122 | 0.033 | 0.008 | 0.004 |
|  | 0.375 | 0.178 | 0.070 | 0.005 | 0.009 | 0.122 | 0.032 | 0.004 | 0.002 | 0.167 | 0.059 | 0.008 | 0.004 | 0.112 | 0.028 | 0.009 | 0.004 | 0.123 | 0.033 | 0.008 | 0.004 |
|  | 0.5 | 0.180 | 0.073 | 0.008 | 0.014 | 0.123 | 0.032 | 0.007 | 0.003 | 0.170 | 0.061 | 0.009 | 0.006 | 0.114 | 0.029 | 0.005 | 0.002 | 0.125 | 0.034 | 0.011 | 0.006 |
|  | mean | 0.174 | 0.067 | 0.010 | 0.013 | 0.124 | 0.032 | 0.006 | 0.004 | 0.170 | 0.061 | 0.008 | 0.006 | 0.111 | 0.027 | 0.008 | 0.003 | 0.125 | 0.034 | 0.009 | 0.005 |
| national illness | 0.125 | 0.612 | 0.840 | 0.122 | 0.477 | 0.521 | 0.722 | 0.157 | 0.597 | 0.619 | 0.916 | 0.164 | 0.660 | 0.454 | 0.590 | 0.129 | 0.522 | 0.443 | 0.588 | 0.145 | 0.541 |
|  | 0.25 | 0.563 | 0.662 | 0.066 | 0.205 | 0.457 | 0.509 | 0.078 | 0.283 | 0.552 | 0.674 | 0.069 | 0.284 | 0.425 | 0.449 | 0.060 | 0.246 | 0.403 | 0.430 | 0.080 | 0.255 |
|  | 0.375 | 0.512 | 0.545 | 0.039 | 0.148 | 0.408 | 0.403 | 0.056 | 0.204 | 0.501 | 0.547 | 0.054 | 0.208 | 0.407 | 0.383 | 0.036 | 0.177 | 0.364 | 0.348 | 0.057 | 0.183 |
|  | 0.5 | 0.521 | 0.544 | 0.019 | 0.084 | 0.406 | 0.374 | 0.031 | 0.139 | 0.516 | 0.538 | 0.032 | 0.147 | 0.407 | 0.361 | 0.028 | 0.119 | 0.360 | 0.320 | 0.040 | 0.130 |
|  | mean | 0.552 | 0.648 | 0.078 | 0.279 | 0.448 | 0.502 | 0.098 | 0.353 | 0.547 | 0.669 | 0.098 | 0.383 | 0.423 | 0.446 | 0.071 | 0.297 | 0.393 | 0.421 | 0.089 | 0.312 |
| traffic | 0.125 | 0.880 | 1.647 | 0.018 | 0.052 | 0.559 | 0.915 | 0.023 | 0.060 | 0.860 | 1.572 | 0.015 | 0.028 | 0.609 | 0.950 | 0.027 | 0.068 | 0.503 | 0.803 | 0.025 | 0.068 |
|  | 0.25 | 0.887 | 1.660 | 0.015 | 0.032 | 0.550 | 0.909 | 0.008 | 0.054 | 0.871 | 1.589 | 0.016 | 0.018 | 0.602 | 0.936 | 0.023 | 0.065 | 0.492 | 0.793 | 0.014 | 0.056 |
|  | 0.375 | 0.874 | 1.620 | 0.011 | 0.034 | 0.548 | 0.901 | 0.012 | 0.052 | 0.860 | 1.553 | 0.011 | 0.028 | 0.603 | 0.931 | 0.015 | 0.043 | 0.487 | 0.780 | 0.006 | 0.044 |
|  | 0.5 | 0.869 | 1.603 | 0.012 | 0.037 | 0.549 | 0.898 | 0.013 | 0.046 | 0.854 | 1.534 | 0.012 | 0.035 | 0.608 | 0.943 | 0.007 | 0.021 | 0.487 | 0.783 | 0.006 | 0.035 |
|  | mean | 0.878 | 1.633 | 0.015 | 0.043 | 0.552 | 0.906 | 0.015 | 0.049 | 0.861 | 1.562 | 0.014 | 0.033 | 0.606 | 0.940 | 0.018 | 0.049 | 0.492 | 0.790 | 0.015 | 0.049 |
| weather | 0.125 | 0.175 | 0.121 | 0.004 | 0.007 | 0.134 | 0.082 | 0.003 | 0.006 | 0.171 | 0.108 | 0.004 | 0.005 | 0.115 | 0.071 | 0.005 | 0.009 | 0.130 | 0.078 | 0.004 | 0.007 |
|  | 0.25 | 0.176 | 0.120 | 0.002 | 0.005 | 0.134 | 0.081 | 0.005 | 0.004 | 0.169 | 0.105 | 0.006 | 0.007 | 0.116 | 0.073 | 0.002 | 0.003 | 0.128 | 0.077 | 0.004 | 0.004 |
|  | 0.375 | 0.177 | 0.121 | 0.001 | 0.002 | 0.134 | 0.083 | 0.005 | 0.007 | 0.168 | 0.105 | 0.005 | 0.007 | 0.117 | 0.076 | 0.003 | 0.006 | 0.129 | 0.080 | 0.005 | 0.007 |
|  | 0.5 | 0.176 | 0.119 | 0.002 | 0.003 | 0.135 | 0.084 | 0.004 | 0.007 | 0.170 | 0.107 | 0.004 | 0.006 | 0.118 | 0.076 | 0.003 | 0.006 | 0.130 | 0.081 | 0.004 | 0.007 |
|  | mean | 0.176 | 0.120 | 0.002 | 0.004 | 0.134 | 0.082 | 0.004 | 0.006 | 0.170 | 0.107 | 0.005 | 0.006 | 0.117 | 0.074 | 0.003 | 0.006 | 0.129 | 0.079 | 0.004 | 0.006 |

Table 4: **Zero-shot imputation performance of `MOMENT`.** Results averaged across five runs with different masking seeds. This table presents the Model Performance Metrics (Mean Absolute Error and Mean Squared Error) on a subset of the Time Series Pile (Goswami et al., 2024; Zhou et al., 2021). The model names include: "Vanilla" `MOMENT-Large` without any pruning, "Block 1-3" for cases where only one block is pruned, and "All" for all three blocks being pruned.

| Dataset | Horizon | Vanilla | | All Pruned | |
|---------|---------|-----|-----|-----|-----|
| | | MAE | MSE | MAE | MSE |
| Exchange | 96 | **0.236** | **0.109** | 0.240 | 0.113 |
| | 192 | **0.333** | **0.215** | 0.335 | 0.218 |
| | 336 | 0.474 | 0.417 | **0.460** | **0.394** |
| | 720 | **0.757** | **1.003** | 0.780 | 1.066 |
| ETTh1 | 96 | **0.409** | **0.385** | **0.409** | 0.388 |
| | 192 | **0.426** | **0.411** | **0.426** | 0.414 |
| | 336 | **0.437** | **0.423** | **0.437** | 0.424 |
| | 720 | **0.464** | **0.443** | 0.470 | 0.460 |
| ETTh2 | 96 | **0.346** | **0.287** | 0.351 | 0.296 |
| | 192 | **0.386** | **0.350** | 0.389 | 0.356 |
| | 336 | **0.409** | **0.370** | 0.413 | 0.382 |
| | 720 | 0.440 | **0.404** | **0.439** | **0.404** |
| ETTm1 | 96 | 0.347 | **0.290** | **0.346** | **0.29** |
| | 192 | 0.373 | 0.330 | **0.369** | **0.326** |
| | 336 | **0.386** | **0.352** | **0.386** | 0.354 |
| | 720 | **0.420** | **0.409** | 0.422 | 0.414 |
| ETTm2 | 96 | **0.260** | **0.171** | 0.262 | 0.173 |
| | 192 | **0.300** | **0.231** | 0.304 | 0.236 |
| | 336 | **0.337** | **0.287** | 0.343 | 0.294 |
| | 720 | **0.392** | **0.372** | 0.395 | **0.372** |
| ILI | 24 | 1.307 | 3.260 | **1.219** | **2.981** |
| | 36 | 1.347 | 3.516 | **1.286** | **3.209** |
| | 48 | 1.405 | 3.828 | **1.374** | **3.479** |
| | 60 | 1.428 | 3.989 | **1.387** | **3.602** |
| Weather | 96 | **0.209** | 0.153 | **0.209** | **0.152** |
| | 192 | **0.248** | **0.197** | **0.248** | 0.198 |
| | 336 | **0.285** | **0.246** | 0.287 | 0.247 |
| | 720 | **0.337** | **0.316** | **0.337** | 0.317 |

Table 5: **Fine-tuned forecasting performance of MOMENT.** This table presents model performance metrics (Mean Absolute Error and Mean Squared Error) on a subset of the Time Series Pile (Goswami et al., 2024; Zhou et al., 2021). Metrics are presented for MOMENT-Large without any pruning (Vanilla) and for all blocks pruned (All). Results are gathered from the best performance on the test set across 3 epochs of training with a batch size 64 and learning rate of 0.0001.

| Method | Pruned Part | Encoder Sparsity (%) | Model Sparsity (%) | Estimated Model Size (MB) | Average Time (ms) | Standard Deviation (ms) |
|---|---|---|---|---|---|---|
| MOMENT | None | 0.00 | 0.00 | 1301.76 | 20.88 | 0.42 |
| | Block 1 | 12.50 | 11.29 | 1154.76 | 19.42 | 0.27 |
| | Block 2 | 33.33 | 30.11 | 909.76 | 19.71 | 0.31 |
| | Block 3 | 12.50 | 11.29 | 1154.76 | 19.56 | 0.29 |
| | All Blocks | 58.33 | 52.70 | 615.76 | 19.82 | 0.43 |
| Chronos-T5 | None | 0.00 | 0.00 | 2704.48 | 59.95 | 0.26 |
| | Block 1 | 8.33 | 3.55 | 2608.48 | 58.22 | 0.58 |
| | Block 2 | 12.50 | 5.32 | 2560.48 | 56.80 | 0.78 |
| | Block 3 | 8.33 | 3.55 | 2608.48 | 56.94 | 0.27 |
| | Block 4 | 25.00 | 10.65 | 2416.48 | 57.55 | 0.84 |
| | All Blocks | 54.17 | 23.07 | 2080.48 | 56.82 | 0.58 |

Table 6: **Inference performance under various pruning configurations.** This table presents the inference performance metrics of the MOMENT-Large and Chronos-Large models with different pruning configurations. Results are presented for MOMENT-Large and Chronos without any pruning (None), when individual blocks pruned Block i, and when all blocks are pruned (All Blocks). Inference time estimation was performed by aggregating times from 100 passes of a one-batch, one-channel sample of length 512.

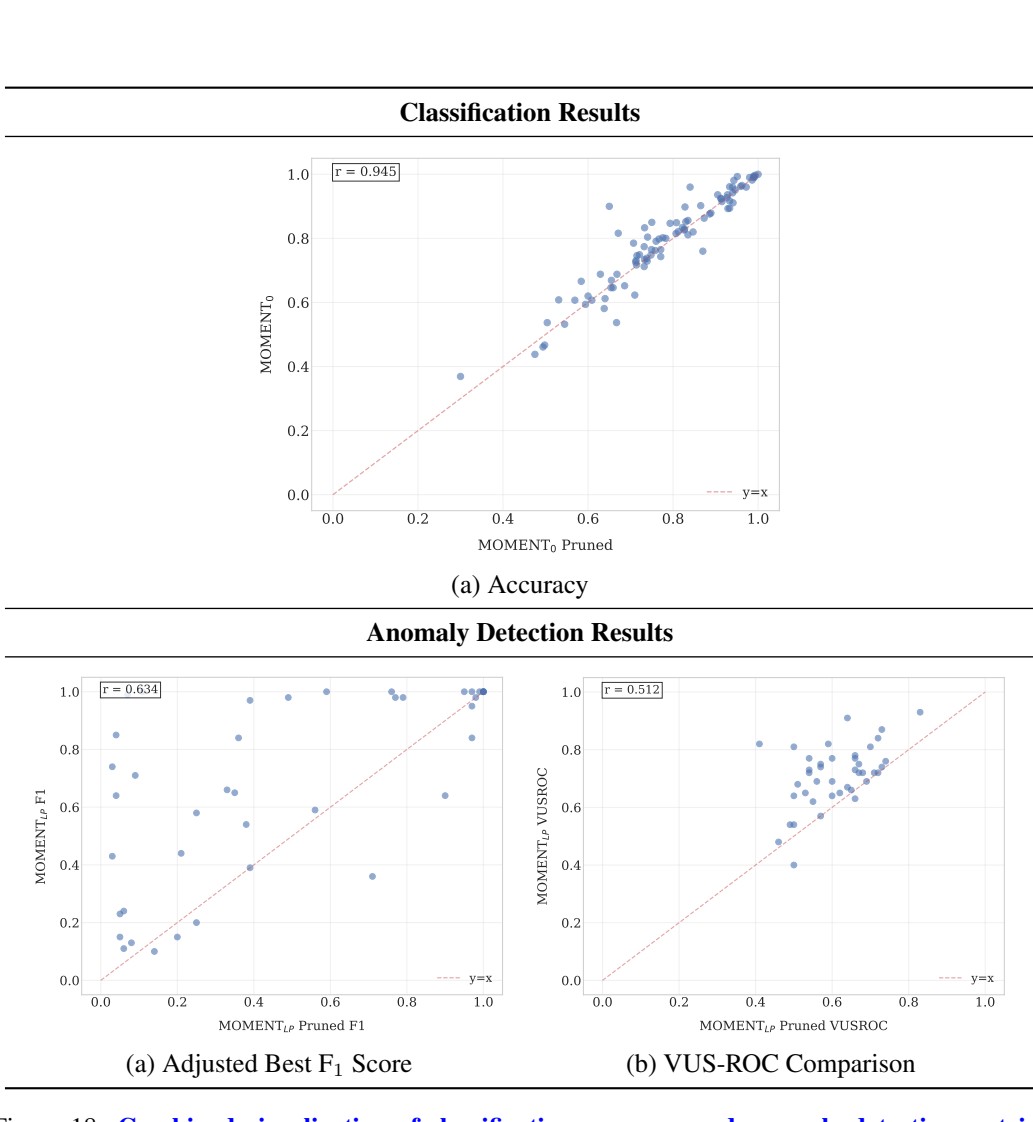

Figure 18: **Combined visualization of classification accuracy and anomaly detection metrics** (F1 Score and VUS-ROC) comparing pruned and non-pruned models. Classification results are based on 91 UCR datasets (MOMENT$_0$ and MOMENT$_{0\_pruned}$), where the mean accuracy is 79.4% and 78.1% respectively (see Table 9). Anomaly detection uses a subset of 44 datasets from the UCR Anomaly Archive (MOMENT$_{LP}$ and MOMENT$_{LP\_pruned}$). See Tables 10 and 8 for detailed results.

| Statistic Category | Metric | Mean Difference | Standard Deviation | Original Better | Pruned Better | Equal | Pearson Correlation Coefficient |
|---|---|---|---|---|---|---|---|
| **Classification** | Accuracy | 0.0128 | 0.0489 | 54 | 28 | 9 | 0.945 |
| **Anomaly Detection** | Adjusted Best $F_1$ | 0.2117 | 0.3026 | 28 | 7 | 6 | 0.634 |
| | VUS-ROC | 0.1007 | 0.0995 | 36 | 2 | 3 | 0.512 |

Table 7: **Summary of differences between MOMENT$_{pruned}$ (all blocks pruned) and MOMENT**. Detailed results are provided in Tables 8 and 10. For classification, we observed a slight deterioration in performance, whereas for anomaly detection the deterioration was more severe.

| | | | Adj. Best F1 | | | | | | | | VUSROC | | | | |
|---|---|---|---|---|---|---|---|---|---|---|---|---|---|---|
| Model name / Dataset name | AnomalyNearestNeighbors | AnomalyTransformer | MOMENT$_{LP}$ | MOMENT$_{LP\_pruned}$ | DGHL | GPT4TS | TimesNet | AnomalyNearestNeighbors | AnomalyTransformer | MOMENT$_{LP}$ | MOMENT$_{LP\_pruned}$ | DGHL | GPT4TS | TimesNet |
| 1sddb40 | 0.720 | 0.030 | 0.540 | 0.380 | 0.390 | 0.190 | 0.680 | 0.680 | 0.640 | 0.750 | 0.570 | 0.640 | 0.660 | 0.720 |
| BIDMC1 | 1.000 | 0.990 | 1.000 | 0.950 | 1.000 | 1.000 | 1.000 | 0.660 | 0.690 | 0.650 | 0.620 | 0.720 | 0.630 | 0.740 |
| CHARISfive | 0.090 | 0.010 | 0.130 | 0.080 | 0.020 | 0.020 | 0.080 | 0.830 | 0.360 | 0.400 | 0.500 | 0.510 | 0.450 | 0.460 |
| CHARISten | 0.020 | 0.020 | 0.110 | 0.060 | 0.040 | 0.010 | 0.030 | 0.520 | 0.430 | 0.540 | 0.500 | 0.520 | 0.510 | 0.530 |
| CIMIS44AirTemperature3 | 1.000 | 0.060 | 0.980 | 0.490 | 0.500 | 0.180 | 0.470 | 0.860 | 0.640 | 0.750 | 0.670 | 0.740 | 0.620 | 0.740 |
| CIMIS44AirTemperature5 | 0.990 | 0.390 | 0.990 | 0.070 | 0.960 | 0.200 | 0.710 | 0.900 | 0.780 | 0.810 | 0.500 | 0.920 | 0.560 | 0.720 |
| ECG2 | 0.360 | 1.000 | 1.000 | 1.000 | 0.620 | 0.900 | 1.000 | 0.840 | 0.830 | 0.840 | 0.720 | 0.630 | 0.780 | 0.600 |
| ECG3 | 1.000 | 0.360 | 0.980 | 0.770 | 0.800 | 0.840 | 0.480 | 0.760 | 0.540 | 0.770 | 0.540 | 0.680 | 0.450 | 0.610 |
| Fantasia | 0.770 | 0.750 | 0.950 | 0.970 | 0.660 | 0.870 | 0.550 | 0.610 | 0.730 | 0.640 | 0.500 | 0.710 | 0.650 | 0.610 |
| GP711MarkerLFM5z4 | 1.000 | 0.930 | 1.000 | 0.760 | 0.500 | 0.640 | 0.950 | 0.790 | 0.540 | 0.730 | 0.540 | 0.600 | 0.620 | 0.720 |
| GP711MarkerLFM5z5 | 1.000 | 0.760 | 0.970 | 0.390 | 0.310 | 0.480 | 0.900 | 0.980 | 0.690 | 0.720 | 0.670 | 0.520 | 0.630 | 0.840 |
| InternalBleeding5 | 0.910 | 0.940 | 1.000 | 0.990 | 1.000 | 0.920 | 1.000 | 0.880 | 0.460 | 0.690 | 0.690 | 0.760 | 0.630 | 0.940 |
| Italianpowerdemand | 0.060 | 0.010 | 0.740 | 0.030 | 0.590 | 0.010 | 0.440 | 0.630 | 0.450 | 0.770 | 0.600 | 0.700 | 0.480 | 0.710 |
| Lab2Cmac011215EPG5 | 0.460 | 0.990 | 0.980 | 0.980 | 0.340 | 0.600 | 0.990 | 0.760 | 0.770 | 0.630 | 0.660 | 0.710 | 0.640 | 0.610 |
| Lab2Cmac011215EPG6 | 0.240 | 0.410 | 0.100 | 0.140 | 0.260 | 0.100 | 0.170 | 0.600 | 0.700 | 0.480 | 0.460 | 0.600 | 0.520 | 0.450 |
| MesoplodonDensirostris | 1.000 | 1.000 | 0.840 | 0.970 | 0.790 | 1.000 | 1.000 | 0.780 | 0.850 | 0.720 | 0.710 | 0.740 | 0.690 | 0.790 |
| PowerDemand1 | 0.800 | 0.870 | 0.440 | 0.210 | 0.490 | 0.760 | 0.950 | 0.800 | 0.720 | 0.540 | 0.490 | 0.530 | 0.600 | 0.750 |
| TkeepFirstMARS | 0.400 | 0.010 | 0.150 | 0.200 | 0.020 | 0.020 | 0.230 | 0.510 | 0.520 | 0.760 | 0.740 | 0.460 | 0.500 | 0.790 |
| TkeepSecondMARS | 0.950 | 0.830 | 1.000 | 1.000 | 0.160 | 0.120 | 0.950 | 0.750 | 0.720 | 0.910 | 0.640 | 0.970 | 0.810 | 0.980 |
| WalkingAcceleration5 | 0.950 | 0.990 | 1.000 | 1.000 | 0.910 | 0.870 | 0.930 | 0.910 | 0.940 | 0.870 | 0.730 | 0.930 | 0.910 | 0.850 |
| apneaecg | 0.360 | 0.400 | 0.200 | 0.250 | 0.250 | 0.310 | 0.260 | 0.700 | 0.580 | 0.690 | 0.560 | 0.590 | 0.580 | 0.760 |
| apneaecg2 | 1.000 | 0.650 | 1.000 | 0.970 | 1.000 | 1.000 | 0.650 | 0.760 | 0.790 | 0.740 | 0.570 | 0.730 | 0.650 | 0.610 |
| gait1 | 1.000 | 0.180 | 0.360 | 0.710 | 0.070 | 0.410 | 0.520 | 0.640 | 0.630 | 0.570 | 0.570 | 0.600 | 0.580 | 0.600 |
| gaitHunt1 | 0.020 | 0.080 | 0.430 | 0.030 | 0.020 | 0.100 | 0.300 | 0.570 | 0.810 | 0.680 | 0.510 | 0.570 | 0.710 | 0.840 |
| insectEPG2 | 0.710 | 0.120 | 0.230 | 0.050 | 0.140 | 0.810 | 0.960 | 0.760 | 0.650 | 0.820 | 0.410 | 0.650 | 0.560 | 0.730 |
| insectEPG4 | 0.650 | 0.980 | 1.000 | 0.110 | 0.460 | 0.210 | 0.850 | 0.760 | 0.690 | 0.720 | 0.540 | 0.730 | 0.490 | 0.650 |
| ltstdbs30791AS | 1.000 | 1.000 | 1.000 | 1.000 | 1.000 | 1.000 | 1.000 | 0.760 | 0.780 | 0.810 | 0.700 | 0.770 | 0.740 | 0.670 |
| mit14046longtermecg | 0.600 | 0.450 | 0.590 | 0.560 | 0.530 | 0.580 | 0.600 | 0.720 | 0.790 | 0.660 | 0.650 | 0.640 | 0.610 | 0.840 |
| park3m | 0.550 | 0.150 | 0.640 | 0.900 | 0.200 | 0.630 | 0.930 | 0.730 | 0.630 | 0.780 | 0.660 | 0.650 | 0.540 | 0.780 |
| qtdbSel1005V | 0.390 | 0.410 | 0.650 | 0.350 | 0.400 | 0.390 | 0.530 | 0.550 | 0.520 | 0.640 | 0.600 | 0.490 | 0.610 | 0.540 |
| qtdbSel100MLII | 0.500 | 0.420 | 0.840 | 0.360 | 0.410 | 0.600 | 0.870 | 0.540 | 0.620 | 0.620 | 0.550 | 0.590 | 0.580 | 0.650 |
| resperation1 | 0.020 | 0.000 | 0.150 | 0.050 | 0.030 | 0.010 | 0.030 | 0.620 | 0.750 | 0.670 | 0.640 | 0.740 | 0.470 | 0.670 |
| s20101mML2 | 0.130 | 0.690 | 0.710 | 0.090 | 0.150 | 0.050 | 0.080 | 0.730 | 0.640 | 0.720 | 0.720 | 0.690 | 0.640 | 0.690 |
| sddb49 | 0.360 | 0.890 | 1.000 | 0.590 | 0.880 | 0.940 | 1.000 | 0.690 | 0.660 | 0.730 | 0.660 | 0.740 | 0.580 | 0.680 |
| sel840mECG1 | 0.350 | 0.160 | 0.660 | 0.330 | 0.280 | 0.210 | 0.360 | 0.740 | 0.620 | 0.720 | 0.680 | 0.870 | 0.650 | 0.600 |
| sel840mECG2 | 0.150 | 0.150 | 0.390 | 0.390 | 0.320 | 0.280 | 0.210 | 0.680 | 0.590 | 0.690 | 0.600 | 0.490 | 0.520 | 0.520 |
| tilt12744mtable | 0.060 | 0.070 | 0.240 | 0.060 | 0.100 | 0.000 | 0.030 | 0.690 | 0.480 | 0.740 | 0.730 | 0.660 | 0.510 | 0.640 |
| tilt12754table | 0.130 | 0.230 | 0.640 | 0.040 | 0.040 | 0.060 | 0.050 | 0.730 | 0.600 | 0.820 | 0.590 | 0.790 | 0.550 | 0.750 |
| tiltAPB2 | 0.690 | 0.920 | 0.980 | 0.790 | 0.360 | 0.830 | 0.380 | 0.710 | 0.770 | 0.660 | 0.710 | 0.660 | 0.710 | 0.700 |
| tiltAPB3 | 0.060 | 0.170 | 0.850 | 0.040 | 0.030 | 0.050 | 0.090 | 0.640 | 0.680 | 0.650 | 0.530 | 0.540 | 0.440 | 0.580 |
| weallwalk | 0.500 | 0.000 | 0.580 | 0.250 | 0.070 | 0.130 | 0.170 | 0.820 | 0.730 | 0.930 | 0.830 | 0.860 | 0.870 | 0.850 |

Table 8: **Anomaly detection** using Adjusted Best $F_1$ and VUS-ROC for a subset of 44 datasets sampled from the UCR Anomaly archive. MOMENT$_{LP}$ and pruned (all blocks) MOMENT$_{LP\_pruned}$.

| | MOMENT$_{0\_pruned}$ | MOMENT$_0$ |
|---|---|---|
| Mean | 0.781 | 0.794 |
| Std | 0.146 | 0.148 |
| Min | 0.300 | 0.369 |
| 25% | 0.697 | 0.714 |
| 50% | 0.776 | 0.815 |
| 75% | 0.915 | 0.916 |
| Max | 1.000 | 1.000 |

Table 9: **Statistic: Classification accuracy** of methods across 91 UCR datasets. MOMENT$_0$ without fine-tuning and pruned (all blocks) MOMENT$_{0\_pruned}$. See full table:10

| Dataset | MOMENT$_{0pruned}$ | MOMENT$_0$ | TS2Vec | T-Loss | TNC | TS-TCC | TST | DTW | CNN | Encoder | FCN | MCDNN | MLP | ResNet | t-LeNet | TWIESN |
|---|---|---|---|---|---|---|---|---|---|---|---|---|---|---|---|---|
| GestureMidAirD2 | 0.531 | 0.608 | 0.469 | 0.546 | 0.362 | 0.254 | 0.138 | 0.608 | 0.518 | 0.480 | 0.631 | 0.500 | 0.545 | 0.668 | 0.038 | 0.575 |
| UWaveGestureLibraryX | 0.812 | 0.821 | 0.795 | 0.785 | 0.781 | 0.733 | 0.569 | 0.728 | 0.721 | 0.771 | 0.754 | 0.726 | 0.768 | 0.781 | 0.127 | 0.608 |
| GesturePebbleZ2 | 0.671 | 0.816 | 0.873 | 0.899 | 0.316 | 0.430 | 0.380 | 0.671 | 0.778 | 0.796 | 0.781 | 0.720 | 0.701 | 0.777 | 0.184 | 0.843 |
| ECG5000 | 0.940 | 0.942 | 0.935 | 0.933 | 0.937 | 0.941 | 0.928 | 0.924 | 0.928 | 0.941 | 0.940 | 0.933 | 0.930 | 0.935 | 0.584 | 0.922 |
| OSULeaf | 0.707 | 0.785 | 0.851 | 0.760 | 0.723 | 0.723 | 0.545 | 0.591 | 0.482 | 0.554 | 0.979 | 0.419 | 0.560 | 0.980 | 0.182 | 0.628 |
| MedicalImages | 0.758 | 0.762 | 0.789 | 0.750 | 0.754 | 0.747 | 0.632 | 0.737 | 0.671 | 0.664 | 0.778 | 0.627 | 0.719 | 0.770 | 0.514 | 0.649 |
| Ham | 0.638 | 0.581 | 0.714 | 0.724 | 0.752 | 0.743 | 0.524 | 0.467 | 0.720 | 0.682 | 0.707 | 0.718 | 0.699 | 0.758 | 0.514 | 0.768 |
| DistalPhalanxTW | 0.640 | 0.612 | 0.698 | 0.676 | 0.669 | 0.676 | 0.568 | 0.590 | 0.671 | 0.694 | 0.695 | 0.685 | 0.610 | 0.663 | 0.285 | 0.591 |
| ProximalPhalanxOutlineCorrect | 0.835 | 0.856 | 0.887 | 0.859 | 0.866 | 0.873 | 0.770 | 0.784 | 0.807 | 0.768 | 0.907 | 0.866 | 0.730 | 0.920 | 0.684 | 0.817 |
| FreezerRegularTrain | 0.986 | 0.982 | 0.986 | 0.956 | 0.991 | 0.989 | 0.922 | 0.899 | 0.987 | 0.760 | 0.997 | 0.973 | 0.906 | 0.998 | 0.500 | 0.946 |
| TwoLeadECG | 0.793 | 0.847 | 0.986 | 0.999 | 0.993 | 0.976 | 0.871 | 0.905 | 0.877 | 0.784 | 0.999 | 0.806 | 0.753 | 1.000 | 0.500 | 0.949 |
| GunPointMaleVersusFemale | 0.991 | 0.991 | 1.000 | 0.997 | 0.994 | 0.997 | 1.000 | 0.997 | 0.977 | 0.978 | 0.997 | 0.952 | 0.980 | 0.992 | 0.525 | 0.988 |
| Trace | 1.000 | 1.000 | 1.000 | 0.990 | 1.000 | 1.000 | 1.000 | 1.000 | 0.952 | 0.740 | 1.000 | 0.902 | 0.806 | 1.000 | 0.240 | 0.934 |
| SmoothSubspace | 0.847 | 0.820 | 0.980 | 0.960 | 0.913 | 0.953 | 0.827 | 0.827 | 0.976 | 0.964 | 0.975 | 0.963 | 0.980 | 0.980 | 0.333 | 0.849 |
| MiddlePhalanxTW | 0.545 | 0.532 | 0.584 | 0.591 | 0.571 | 0.610 | 0.506 | 0.551 | 0.597 | 0.583 | 0.562 | 0.536 | 0.495 | 0.286 | 0.569 | |
| SyntheticControl | 0.980 | 0.990 | 0.997 | 0.987 | 1.000 | 0.990 | 0.490 | 0.993 | 0.987 | 0.973 | 0.989 | 0.953 | 0.973 | 0.997 | 0.167 | 0.879 |
| ShapesAll | 0.807 | 0.815 | 0.902 | 0.848 | 0.788 | 0.773 | 0.511 | 0.679 | 0.617 | 0.679 | 0.894 | 0.599 | 0.776 | 0.926 | 0.017 | 0.643 |
| AllGestureWiimoteX | 0.569 | 0.607 | 0.777 | 0.763 | 0.703 | 0.697 | 0.259 | 0.716 | 0.411 | 0.475 | 0.713 | 0.261 | 0.477 | 0.741 | 0.100 | 0.522 |
| Wafer | 0.993 | 0.997 | 0.998 | 0.992 | 0.994 | 0.994 | 0.991 | 0.980 | 0.961 | 0.998 | 0.997 | 0.992 | 0.996 | 0.998 | 0.892 | 0.916 |
| FaceFour | 0.830 | 0.852 | 0.932 | 0.920 | 0.659 | 0.773 | 0.511 | 0.830 | 0.905 | 0.852 | 0.930 | 0.711 | 0.836 | 0.955 | 0.295 | 0.857 |
| CricketX | 0.721 | 0.749 | 0.782 | 0.713 | 0.623 | 0.731 | 0.385 | 0.754 | 0.535 | 0.644 | 0.794 | 0.513 | 0.591 | 0.799 | 0.074 | 0.627 |
| DistalPhalanxOutlineCorrect | 0.714 | 0.717 | 0.761 | 0.775 | 0.754 | 0.754 | 0.728 | 0.717 | 0.772 | 0.724 | 0.760 | 0.759 | 0.727 | 0.770 | 0.583 | 0.711 |
| ChlorineConcentration | 0.771 | 0.765 | 0.832 | 0.749 | 0.760 | 0.753 | 0.562 | 0.648 | 0.608 | 0.583 | 0.817 | 0.662 | 0.800 | 0.853 | 0.533 | 0.554 |
| Chinatown | 0.962 | 0.965 | 0.965 | 0.951 | 0.977 | 0.983 | 0.936 | 0.957 | 0.977 | 0.966 | 0.980 | 0.945 | 0.872 | 0.978 | 0.726 | 0.825 |
| GestureMidAirD1 | 0.654 | 0.646 | 0.608 | 0.608 | 0.431 | 0.369 | 0.208 | 0.569 | 0.534 | 0.528 | 0.695 | 0.518 | 0.575 | 0.698 | 0.038 | 0.549 |
| MiddlePhalanxOutlineAgeGroup | 0.494 | 0.461 | 0.636 | 0.656 | 0.643 | 0.630 | 0.617 | 0.500 | 0.534 | 0.577 | 0.535 | 0.558 | 0.522 | 0.545 | 0.571 | 0.578 |
| UMD | 0.951 | 0.993 | 1.000 | 0.993 | 0.993 | 0.986 | 0.910 | 0.993 | 0.960 | 0.771 | 0.988 | 0.842 | 0.949 | 0.990 | 0.333 | 0.835 |
| Crop | 0.733 | 0.734 | 0.756 | 0.722 | 0.738 | 0.742 | 0.710 | 0.665 | 0.670 | 0.760 | 0.738 | 0.687 | 0.618 | 0.743 | 0.042 | 0.489 |
| GesturePebbleZ1 | 0.808 | 0.849 | 0.930 | 0.919 | 0.378 | 0.395 | 0.500 | 0.791 | 0.844 | 0.821 | 0.880 | 0.769 | 0.792 | 0.901 | 0.163 | 0.840 |
| WordSynonyms | 0.668 | 0.688 | 0.676 | 0.691 | 0.630 | 0.531 | 0.422 | 0.649 | 0.568 | 0.557 | 0.561 | 0.470 | 0.599 | 0.617 | 0.219 | 0.506 |
| ArrowHead | 0.771 | 0.743 | 0.857 | 0.766 | 0.703 | 0.737 | 0.771 | 0.703 | 0.717 | 0.630 | 0.843 | 0.678 | 0.784 | 0.838 | 0.303 | 0.689 |
| Wine | 0.667 | 0.537 | 0.870 | 0.815 | 0.759 | 0.778 | 0.500 | 0.574 | 0.519 | 0.556 | 0.611 | 0.500 | 0.541 | 0.722 | 0.500 | 0.744 |
| Coffee | 0.929 | 0.893 | 1.000 | 1.000 | 1.000 | 1.000 | 0.821 | 1.000 | 1.000 | 0.886 | 1.000 | 0.979 | 0.993 | 1.000 | 0.507 | 0.979 |
| Earthquakes | 0.748 | 0.748 | 0.748 | 0.748 | 0.748 | 0.748 | 0.719 | 0.709 | 0.740 | 0.725 | 0.748 | 0.748 | 0.727 | 0.712 | 0.748 | 0.748 |
| Herring | 0.594 | 0.594 | 0.641 | 0.594 | 0.594 | 0.594 | 0.594 | 0.531 | 0.531 | 0.512 | 0.644 | 0.572 | 0.491 | 0.600 | 0.594 | 0.625 |
| Beef | 0.733 | 0.833 | 0.767 | 0.667 | 0.733 | 0.600 | 0.500 | 0.633 | 0.767 | 0.707 | 0.680 | 0.507 | 0.713 | 0.753 | 0.200 | 0.527 |
| MiddlePhalanxOutlineCorrect | 0.498 | 0.467 | 0.838 | 0.825 | 0.818 | 0.818 | 0.753 | 0.698 | 0.744 | 0.752 | 0.795 | 0.796 | 0.755 | 0.826 | 0.570 | 0.743 |
| ECGFiveDays | 0.740 | 0.804 | 1.000 | 1.000 | 0.999 | 0.878 | 0.763 | 0.768 | 0.874 | 0.842 | 0.985 | 0.800 | 0.973 | 0.966 | 0.497 | 0.723 |
| Yoga | 0.822 | 0.834 | 0.887 | 0.837 | 0.812 | 0.791 | 0.830 | 0.837 | 0.786 | 0.753 | 0.837 | 0.741 | 0.856 | 0.867 | 0.536 | 0.626 |
| Adiac | 0.629 | 0.688 | 0.762 | 0.675 | 0.726 | 0.767 | 0.550 | 0.604 | 0.393 | 0.318 | 0.841 | 0.620 | 0.391 | 0.833 | 0.023 | 0.428 |
| MoteStrain | 0.732 | 0.774 | 0.861 | 0.851 | 0.825 | 0.843 | 0.768 | 0.835 | 0.885 | 0.872 | 0.936 | 0.691 | 0.855 | 0.924 | 0.539 | 0.809 |
| Strawberry | 0.946 | 0.951 | 0.962 | 0.954 | 0.951 | 0.965 | 0.916 | 0.941 | 0.952 | 0.959 | 0.975 | 0.958 | 0.959 | 0.980 | 0.643 | 0.911 |
| InsectWingbeatSound | 0.609 | 0.607 | 0.630 | 0.597 | 0.549 | 0.415 | 0.266 | 0.355 | 0.585 | 0.630 | 0.392 | 0.587 | 0.604 | 0.499 | 0.091 | 0.435 |
| DodgerLoopWeekend | 0.826 | 0.826 | 0.964 | NaN | NaN | NaN | 0.732 | 0.949 | 0.974 | 0.983 | 0.904 | 0.978 | 0.978 | 0.952 | 0.739 | 0.954 |
| Meat | 0.933 | 0.917 | 0.950 | 0.950 | 0.917 | 0.883 | 0.900 | 0.933 | 0.913 | 0.787 | 0.803 | 0.787 | 0.893 | 0.990 | 0.333 | 0.970 |
| MelbournePedestrian | 0.886 | 0.876 | 0.959 | 0.944 | 0.942 | 0.949 | 0.741 | 0.791 | 0.813 | 0.884 | 0.912 | 0.840 | 0.863 | 0.909 | 0.100 | 0.730 |
| FaceAll | 0.760 | 0.791 | 0.771 | 0.786 | 0.766 | 0.813 | 0.504 | 0.808 | 0.774 | 0.794 | 0.938 | 0.720 | 0.794 | 0.867 | 0.080 | 0.673 |
| FacesUCR | 0.835 | 0.811 | 0.924 | 0.884 | 0.789 | 0.863 | 0.543 | 0.905 | 0.873 | 0.867 | 0.943 | 0.775 | 0.831 | 0.954 | 0.143 | 0.641 |
| AllGestureWiimoteY | 0.584 | 0.666 | 0.793 | 0.726 | 0.699 | 0.741 | 0.423 | 0.729 | 0.479 | 0.509 | 0.744 | 0.420 | 0.571 | 0.794 | 0.100 | 0.600 |
| ShakeGestureWiimoteZ | 0.840 | 0.960 | 0.940 | 0.920 | 0.820 | 0.860 | 0.760 | 0.860 | 0.580 | 0.756 | 0.884 | 0.516 | 0.548 | 0.880 | 0.100 | 0.864 |
| BME | 0.940 | 0.960 | 0.993 | 0.993 | 0.973 | 0.933 | 0.760 | 0.900 | 0.947 | 0.827 | 0.836 | 0.896 | 0.905 | 0.999 | 0.333 | 0.819 |
| FordB | 0.767 | 0.798 | 0.794 | 0.793 | 0.733 | 0.815 | 0.507 | 0.620 | 0.749 | 0.777 | 0.772 | 0.698 | 0.707 | 0.813 | 0.503 | 0.512 |
| Fish | 0.783 | 0.800 | 0.926 | 0.891 | 0.817 | 0.817 | 0.720 | 0.823 | 0.855 | 0.734 | 0.961 | 0.720 | 0.848 | 0.981 | 0.126 | 0.878 |
| SonyAIBORobotSurface2 | 0.827 | 0.829 | 0.871 | 0.889 | 0.834 | 0.907 | 0.745 | 0.831 | 0.831 | 0.844 | 0.980 | 0.804 | 0.831 | 0.975 | 0.617 | 0.635 |
| FiftyWords | 0.776 | 0.802 | 0.771 | 0.732 | 0.653 | 0.653 | 0.525 | 0.690 | 0.624 | 0.658 | 0.646 | 0.611 | 0.708 | 0.740 | 0.125 | 0.518 |
| ToeSegmentation1 | 0.912 | 0.925 | 0.917 | 0.939 | 0.864 | 0.930 | 0.807 | 0.772 | 0.598 | 0.706 | 0.961 | 0.559 | 0.589 | 0.957 | 0.526 | 0.882 |
| FreezerSmallTrain | 0.865 | 0.902 | 0.870 | 0.933 | 0.982 | 0.979 | 0.920 | 0.753 | 0.739 | 0.676 | 0.683 | 0.688 | 0.686 | 0.832 | 0.500 | 0.917 |
| TwoPatterns | 0.989 | 0.994 | 1.000 | 0.999 | 1.000 | 0.999 | 0.466 | 1.000 | 0.991 | 1.000 | 0.870 | 0.976 | 0.948 | 1.000 | 0.259 | 0.875 |
| ShapeletSim | 0.933 | 0.961 | 1.000 | 0.672 | 0.589 | 0.683 | 0.489 | 0.650 | 0.497 | 0.510 | 0.706 | 0.498 | 0.513 | 0.782 | 0.500 | 0.546 |
| Plane | 0.990 | 0.990 | 1.000 | 0.990 | 1.000 | 1.000 | 0.933 | 1.000 | 0.962 | 0.964 | 1.000 | 0.952 | 0.977 | 1.000 | 0.143 | 1.000 |
| GestureMidAirD3 | 0.300 | 0.369 | 0.292 | 0.285 | 0.292 | 0.177 | 0.154 | 0.323 | 0.317 | 0.368 | 0.326 | 0.278 | 0.382 | 0.340 | 0.038 | 0.275 |
| DiatomSizeReduction | 0.889 | 0.879 | 0.984 | 0.984 | 0.993 | 0.977 | 0.961 | 0.967 | 0.954 | 0.880 | 0.346 | 0.646 | 0.909 | 0.301 | 0.301 | 0.914 |
| CricketZ | 0.713 | 0.731 | 0.792 | 0.708 | 0.682 | 0.713 | 0.403 | 0.754 | 0.501 | 0.651 | 0.810 | 0.484 | 0.629 | 0.809 | 0.062 | 0.643 |
| Lightning7 | 0.712 | 0.726 | 0.863 | 0.795 | 0.767 | 0.685 | 0.411 | 0.726 | 0.647 | 0.696 | 0.825 | 0.559 | 0.616 | 0.827 | 0.260 | 0.608 |
| UWaveGestureLibraryY | 0.738 | 0.738 | 0.719 | 0.710 | 0.697 | 0.641 | 0.348 | 0.634 | 0.626 | 0.676 | 0.642 | 0.639 | 0.699 | 0.666 | 0.121 | 0.497 |
| GunPointAgeSpan | 0.959 | 0.962 | 0.987 | 0.994 | 0.984 | 0.994 | 0.991 | 0.918 | 0.912 | 0.890 | 0.996 | 0.887 | 0.934 | 0.997 | 0.494 | 0.965 |
| DistalPhalanxOutlineAgeGroup | 0.655 | 0.669 | 0.727 | 0.727 | 0.741 | 0.755 | 0.741 | 0.770 | 0.758 | 0.761 | 0.718 | 0.729 | 0.647 | 0.718 | 0.433 | 0.705 |
| SwedishLeaf | 0.914 | 0.923 | 0.941 | 0.914 | 0.880 | 0.923 | 0.738 | 0.792 | 0.884 | 0.902 | 0.967 | 0.841 | 0.845 | 0.963 | 0.064 | 0.837 |
| CBF | 0.972 | 0.960 | 1.000 | 0.983 | 0.983 | 0.998 | 0.898 | 0.997 | 0.959 | 0.977 | 0.994 | 0.908 | 0.869 | 0.996 | 0.332 | 0.896 |
| BeetleFly | 0.650 | 0.900 | 0.900 | 0.800 | 0.850 | 0.800 | 1.000 | 0.700 | 0.900 | 0.620 | 0.910 | 0.630 | 0.880 | 0.850 | 0.500 | 0.790 |
| AllGestureWiimoteZ | 0.504 | 0.537 | 0.746 | 0.723 | 0.646 | 0.689 | 0.447 | 0.643 | 0.375 | 0.396 | 0.692 | 0.287 | 0.439 | 0.726 | 0.100 | 0.516 |
| DodgerLoopDay | 0.475 | 0.438 | 0.562 | NaN | NaN | NaN | 0.200 | 0.500 | 0.312 | 0.487 | 0.143 | 0.305 | 0.160 | 0.150 | 0.160 | 0.593 |
| GunPointOldVersusYoung | 0.943 | 0.981 | 1.000 | 1.000 | 1.000 | 1.000 | 1.000 | 0.838 | 0.922 | 0.923 | 0.989 | 0.926 | 0.941 | 0.989 | 0.524 | 0.975 |
| FordA | 0.905 | 0.936 | 0.936 | 0.928 | 0.902 | 0.930 | 0.568 | 0.555 | 0.896 | 0.928 | 0.914 | 0.863 | 0.816 | 0.937 | 0.510 | 0.555 |
| ItalyPowerDemand | 0.941 | 0.911 | 0.925 | 0.954 | 0.928 | 0.955 | 0.845 | 0.950 | 0.954 | 0.964 | 0.963 | 0.966 | 0.953 | 0.962 | 0.499 | 0.871 |
| ProximalPhalanxOutlineAgeGroup | 0.873 | 0.863 | 0.834 | 0.844 | 0.854 | 0.839 | 0.854 | 0.805 | 0.812 | 0.872 | 0.825 | 0.839 | 0.849 | 0.847 | 0.488 | 0.839 |
| GunPoint | 0.927 | 0.927 | 0.980 | 0.980 | 0.967 | 0.993 | 0.827 | 0.907 | 0.948 | 0.784 | 1.000 | 0.907 | 0.907 | 0.991 | 0.493 | 0.989 |
| ProximalPhalanxTW | 0.732 | 0.712 | 0.824 | 0.771 | 0.810 | 0.800 | 0.780 | 0.761 | 0.777 | 0.791 | 0.761 | 0.775 | 0.767 | 0.773 | 0.341 | 0.784 |
| PickupGestureWiimoteZ | 0.600 | 0.620 | 0.820 | 0.740 | 0.620 | 0.600 | 0.240 | 0.660 | 0.608 | 0.496 | 0.744 | 0.412 | 0.604 | 0.704 | 0.100 | 0.616 |
| SonyAIBORobotSurface1 | 0.739 | 0.729 | 0.903 | 0.902 | 0.804 | 0.899 | 0.724 | 0.725 | 0.690 | 0.729 | 0.958 | 0.655 | 0.692 | 0.961 | 0.429 | 0.725 |
| PowerCons | 0.933 | 0.894 | 0.961 | 0.900 | 0.933 | 0.961 | 0.911 | 0.878 | 0.960 | 0.971 | 0.863 | 0.929 | 0.977 | 0.879 | 0.500 | 0.852 |
| PhalangesOutlinesCorrect | 0.686 | 0.652 | 0.809 | 0.784 | 0.787 | 0.804 | 0.773 | 0.728 | 0.799 | 0.745 | 0.818 | 0.795 | 0.756 | 0.845 | 0.613 | 0.656 |
| BirdChicken | 0.750 | 0.850 | 0.800 | 0.850 | 0.750 | 0.650 | 0.650 | 0.750 | 0.710 | 0.510 | 0.940 | 0.540 | 0.740 | 0.880 | 0.500 | 0.620 |
| ToeSegmentation2 | 0.915 | 0.915 | 0.892 | 0.900 | 0.831 | 0.877 | 0.615 | 0.838 | 0.752 | 0.702 | 0.889 | 0.649 | 0.745 | 0.894 | 0.815 | 0.794 |
| CricketY | 0.715 | 0.746 | 0.749 | 0.728 | 0.597 | 0.718 | 0.467 | 0.744 | 0.582 | 0.639 | 0.793 | 0.521 | 0.598 | 0.810 | 0.085 | 0.652 |
| ElectricDevices | 0.659 | 0.646 | 0.721 | 0.707 | 0.700 | 0.686 | 0.676 | 0.602 | 0.686 | 0.702 | 0.706 | 0.653 | 0.593 | 0.728 | 0.242 | 0.605 |
| DodgerLoopGame | 0.710 | 0.623 | 0.841 | NaN | NaN | NaN | 0.696 | 0.877 | 0.816 | 0.810 | 0.768 | 0.877 | 0.865 | 0.710 | 0.478 | 0.716 |
| Fungi | 0.828 | 0.898 | 0.957 | 1.000 | 0.527 | 0.753 | 0.366 | 0.839 | 0.961 | 0.934 | 0.018 | 0.051 | 0.863 | 0.177 | 0.063 | 0.439 |
| Symbols | 0.928 | 0.936 | 0.976 | 0.963 | 0.885 | 0.916 | 0.786 | 0.950 | 0.808 | 0.754 | 0.955 | 0.644 | 0.836 | 0.893 | 0.174 | 0.798 |
| UWaveGestureLibraryZ | 0.749 | 0.765 | 0.770 | 0.757 | 0.721 | 0.690 | 0.655 | 0.658 | 0.630 | 0.684 | 0.727 | 0.645 | 0.697 | 0.749 | 0.121 | 0.573 |
| ECG200 | 0.870 | 0.760 | 0.920 | 0.940 | 0.830 | 0.880 | 0.830 | 0.770 | 0.816 | 0.884 | 0.888 | 0.838 | 0.914 | 0.874 | 0.640 | 0.874 |

Table 10: **Classification accuracy** of methods across 91 UCR datasets. Results are shown for MOMENT$_0$ (without fine-tuning) and its pruned version, MOMENT$_{0\_pruned}$ (all blocks). We report that MOMENT$_0$ achieves 54 wins, 9 ties, and 28 losses compared to MOMENT$_{0\_pruned}$.

