# OpenReview forum: "Exploring Representations and Interventions in Time Series Foundation Models"
_ICLR.cc/2025/Conference — Submitted to ICLR 2025_

### Official Review · Reviewer_MRNR · 2024-10-31

**Soundness:** 3
**Presentation:** 2
**Contribution:** 3
**Rating:** 6
**Confidence:** 3

**Summary:**

In this paper, the author investigates the structure and redundancy of representations across various time series foundation models, examining the self-similarity of model layers within and across different model sizes. This analysis reveals block-like redundancy in the representations, which can be utilized for informed pruning to improve inference speed and efficiency. Additionally, the autjor explores the concepts learned by these models—such as periodicity and trends—and how these can be manipulated through latent space steering to influence model behavior. These findings underscore the value of representational analysis for optimizing models and demonstrate how conceptual steering offers new possibilities for more controlled and efficient time series analysis with TSFMs.

**Strengths:**

The starting point of this article is very meaningful, being at a time when TSFMs are emerging, it is crucial to explore more efficient, interpretable, and controllable models. Additionally, some experimental findings in the article provide guidance for future TSFM architecture design. Moreover, the figures and tables in this paper are also aesthetically pleasing.

**Weaknesses:**

I have no objections to the content of this article. My only concern is that it reads more like a technical report, lacking an overall logical structure. Many figures and text sections appear isolated, which could make it difficult for beginners to read and understand. I suggest the authors expand the Introduction section to provide readers with a more comprehensive overview (currently only half a page) and adjust the layout of the figures to accurately correspond with the text content.

**Questions:**

* **Q1**: I am a bit unclear about the concept of "block" in the article. Does it refer to some adjacent layers? What does it mean to prune Block3 of MOMENT-large at line 426?

* **Q2**: Can you provide a more detailed explanation on how to control the generation of time series? Does this involve modifying some parameters of the model?

---

> ### Author Response · Authors · 2024-11-21
> **Changes in response to your feedback**
>
> Dear Reviewer MRNR,
>
> Thank you very much for your thoughtful and detailed review of our paper. We are glad that you found our contributions to be well-timed and well-motivated, and that they “provide guidance for future TSFM architecture design.” We are delighted that you found the figures and tables in our paper to be “aesthetically pleasing,” as we devoted considerable effort to their design to convey our methods clearly and effectively.
>
> You highlighted two key areas for improvement: (1) expanding the Introduction section to provide readers with a more comprehensive overview, and (2) adjusting the layout of figures to better correspond with the text content. Based on your feedback, we have made several revisions to address these points. For your convenience, we have highlighted these changes in blue ink in the revised manuscript.
>
> 1. **Expanded introduction:** We have revised the Introduction section to provide a more comprehensive overview of our motivation, methods, and experimental results, offering readers better context and a stronger logical structure.
> 2. **Enhanced Subsection Organization in Section 3.2:** Based on your feedback and additional input from Reviewer kpxg, we have restructured Section 3.2 by incorporating `\subsubsections` to enhance its readability and coherence.
> 3. **Adjusted Figure Layout:** We have realigned Figures 5–9 to place them closer to the corresponding text, ensuring a more seamless connection between the figures and their discussion.
>
> ### Questions
> **I am a bit unclear about the concept of "block" in the article. Does it refer to some adjacent layers? What does it mean to prune Block3 of MOMENT-large at line 426?**
> 1.  Thank you for this clarifying question. In our paper, the term "block" refers to groups of consecutive layers within a transformer that exhibit high representational similarity. Consistent with prior work [1], we use "layer" to refer to individual transformer encoder or decoder blocks consistent with prior work, while "block" refers to a higher-level structure made up of multiple such layers that share similar representations [2].
> 2. As an example, consider Figure 1 which illustrates the pairwise similarity between the 24 layers of $\texttt{MOMENT-Large}$. In this figure, lighter colors (yellow) represent higher representational similarity, as measured by Centered Kernel Alignment (CKA) (Kornblith et al., 2019). The identified blocks are outlined with red bounding boxes. For example, Block 1 comprises layers 1–5, Block 2 comprises layers 9–18, and Block 3 comprises layers 19–23.
> 3. Our pruning algorithm is summarized in Algorithm 1, and lines 152–154. Regarding your specific question about pruning Block 3 of $\texttt{MOMENT-Large}$, our pruning method involves retaining the first and last layers of each block while zeroing out the weights of the intermediate layers. This approach preserves the structural integrity of the block while leveraging the skip connections within transformer blocks to ensure that signals and gradients continue to flow through the network. In the case of Block 3, composed of layers 19–23, this means that layers 19 and 23 are retained, while the weights of layers 20, 21, and 22 are zeroed out. We have added a dedicated section on pruning in Appendix C to further clarify this process.
> 4. We identify redundant blocks in a TSFM through visual inspection, which aligns with prior work [2]. Table 1 lists all the identified blocks in $\texttt{MOMENT-Large}$ and $\texttt{Chronos-Large}$. In addition to visual inspection, redundant blocks can also be identified using algorithmic approaches. As an example we outline a simple algorithmic approach in Appendix C.
>
> ### References
> 1. Phang, Jason, Haokun Liu, and Samuel R. Bowman. "Fine-tuned transformers show clusters of similar representations across layers." arXiv preprint arXiv:2109.08406 (2021).
> 2. Nguyen, Thao, Maithra Raghu, and Simon Kornblith. "Do wide and deep networks learn the same things? uncovering how neural network representations vary with width and depth." arXiv preprint arXiv:2010.15327 (2020).

---

> > ### Author Response · Authors · 2024-11-21
> > **Answer to your question**
> >
> > **Can you provide a more detailed explanation on how to control the generation of time series? Does this involve modifying some parameters of the model?**
> > 1. We believe this question refers to the steering experiments illustrated in Figure 9. To control model predictions—forecasting in $\texttt{Chronos}$ or reconstruction in $\texttt{MOMENT}$—we employed the strategy described in Section 3.2.2 (Deriving Steering Matrices for Model Steering). This method allows us to steer model outputs without modifying the model parameters by instead adjusting the hidden activations during inference. As an example, consider that we want to add an upward trend to the model's predictions. We outline the steering process steps below:
> >     1. **Deriving Steering Vectors**: For each layer i in the model, we derive a steering vector $\mathbf{S}_i$, which represents the change that hidden activations in layer $i$ must undergo to induce the desired effect (e.g., an upward trend) in the model's predictions. To compute $\mathbf{S}_i$, we use a set of (synthetic) time series with known trends. For instance, if we have 10 time series, 5 with no trend and 5 with an upward trend, we calculate $\mathbf{S}_i$ as the difference between the median hidden activations of time series with an upward trend and those with no trend at layer $i$.
> >     2. **Constructing a Steering Matrix:** The steering vectors derived for all layers of the model are stacked into a steering matrix. This matrix provides layer-wise and token wise guidance on how to adjust hidden activations to bias the model’s outputs in the desired direction.
> >     3. **Applying Steering During Inference**: During inference, we do not modify the model's parameters. Instead, at each layer $i$, we update the hidden activations by adding the corresponding steering vector $\mathbf{S}_i$. If $\mathbf{h}_i$ denotes the hidden activations of layer $i$, we update its hidden representation as follows: $\mathbf{h}_i \leftarrow \mathbf{h}_i + \lambda \mathbf{S}_i$, where $\lambda$ controls the strength of the intervention. This adjustment biases the activations to produce outputs aligned with the desired trend (e.g., an upward trend in predictions).
> > 2. If the question was instead referring to how we generate (synthetic) time series used to derive steering vectors the steering experiments, then we would like to point to Appendix A where we outline the synthetic data generation process and its hyper-parameters. Figure 10 in the Appendix A shows samples of synthetic time series used in our experiments. As shown in the figure, we use two base patterns (constant and sinusoidal). To generate synthetic datasets, we vary the periodicity $f$, amplitude $a$, intercept $b$, and linear trend $m$.
> >
> > Thank you again for your insightful feedback, which has helped improve the clarity and flow of our paper. We hope that these revisions have addressed your concerns. Please let us know if there are more ways to improve our work. If you feel that we have addressed your concerns, we kindly ask that you reconsider your score in light of the revisions.

---

> > > ### Comment · Reviewer_MRNR · 2024-11-26
> > >
> > > Thanks for the author's reply. They have answered my questions. I will maintain my score (I think 6 is the appropriate score for this paper; the rest is up to AC to decide. Good luck to the author).

---

> ### Author Response · Authors · 2024-11-26
> **Thanks for your feedback!**
>
> Dear Reviewer MRNR,
>
> Thank you for your thoughtful feedback and for engaging with us. We are glad to hear that our response has addressed your concerns. If there are any remaining points or questions that require further clarification or elaboration, we would be more than happy to address them. If possible, we would be grateful for any additional insights into areas where you think our paper could be strengthened or improved in order to merit a higher score. Once again, we greatly appreciate your constructive input, which has helped improve the quality of our paper.

---

### Official Review · Reviewer_QT6A · 2024-11-03

**Soundness:** 3
**Presentation:** 2
**Contribution:** 3
**Rating:** 6
**Confidence:** 4

**Summary:**

This work investigates the internal representations of Time Series Foundation Models (TSFMs), revealing block-like redundancy in their representations. It leverages this redundancy for informed pruning to enhance inference speed and efficiency without compromising accuracy. The study also explores learned concepts like periodicity and trends, demonstrating the potential of latent space steering to manipulate these concepts and influence model behavior, introducing new features to time series signals. The findings highlight the importance of representational analysis for optimizing TSFMs and open new avenues for controlled time series analysis.

**Strengths:**

1 By comparing representational similarity between layers of the model, this work provides a new method for analyzing TSFMs. Systematic probing of TSFMs through representational similarity identifies redundant representations suitable for model pruning.

2 The work effectively reduces model size and improves inference speed while maintaining accuracy. It introduces the ability to steer model predictions along conceptual directions, influencing outputs in targeted ways.

3 The study provides comprehensive analytical experiments, verifying that steering interventions can introduce new features into signals, such as periodicity and trends. Moreover, it is not limited to a single model but spans multiple TSFMs, enhancing the generality of the findings.The use of open-source models and data facilitates community reproduction and further research.

**Weaknesses:**

1 Although a pruning strategy is proposed, the effectiveness of reducing computational resource consumption for extremely large models like TimeGPT has not been fully validated.

2 The selection of the parameter α in concept steering may require fine-tuning, increasing the complexity of model application. Have you explored automated methods for selecting the parameter α, or can you provide guidance on choosing appropriate α values for different dataset and models.

3 The study primarily focuses on forecasting and does not explore the generalization of the findings to other time series tasks such as anomaly detection or classification.

4 The computational overhead of calculating and applying steering matrices in real-time scenarios is not addressed, which could be significant for large-scale applications.

**Questions:**

Refer to weaknesses

---

> ### Author Response · Authors · 2024-11-22
> **Changes in response to your feedback**
>
> Dear Reviewer QT6A,
>
> Thank you so much for your time and thoughtful feedback! Thank you for recognizing that our findings are “not limited to a single model but span multiple TSFMs, enhancing the generality of the findings.” We have put significant thought into your feedback, and carried out multiple experiments to address your concerns.
> 1. **Effectiveness of pruning on large models:** We conducted pruning experiments on two of the largest open-source models: MOMENT-Large (385M parameters) and Chronos-Large (710M parameters) at the time of writing this paper (see Table 8 in [1]). The results from these experiments are summarized in Tables 2–10, and Figure 18 of the revised manuscript. Unfortunately, we were unable to include TimeGPT in these experiments, as it is a closed-source model only accessible via paid API. Consequently, we could not obtain the necessary access for direct pruning experiments on TimeGPT.
> 2. **On selecting the parameter $\alpha$**: We rename $\alpha$ to $\lambda$ in the revised manuscript for clarity. Below we provide some guidance on selecting good values of $\lambda$ and insights into its properties:
>     1. **Selection and Impact of the Steering Strength Parameter $\lambda$:**
>         1. **Optimal Range:** Based on our empirical experiments, we found that the steering strength parameter $\lambda$ is most effective for interventions when its value lies within the interval $[0.1, 2.0]$.
>         2. **Lower Bound Considerations:** Values of $\lambda < 0.1$ often result in insufficient perturbation of the activation patterns, leading to suboptimal intervention effects that may not manifest visibly in the model's output.
>         3. **Upper Bound Effects:** Setting $\lambda > 2.0$ induces excessive perturbations that push activations beyond their typical distribution bounds, potentially resulting in degenerate or semantically meaningless outputs. In the  PCA/latent space visualizations, these cases simply appear as more distant points along the steering direction.
>     2. **Directional Properties:**
>         1. **Reversibility:** Multiplying the steering vector by $-1$ effectively reverses the direction of intervention, enabling bidirectional control (e.g., transforming concept $A \rightarrow B$ into $B \rightarrow A$).
>         2. Example Application: For a steering vector trained to increase signal magnitude, applying its negative counterpart (-$\lambda$ $\mathbf{S}$) produces controlled signal decrease, demonstrating the symmetric nature of the steering operation.
>     3. **Practical Guidelines:**
>         1. **Initial Calibration:** We recommend starting with $\lambda = 1.0$ and adjusting based on the observed intervention strength. In most cases value $\lambda = 1.0$ works well and does not need tuning.
>         2. **Task and Model-Specific Tuning:** If $\lambda = 1.0$ does not yield satisfactory results, the optimal value requires tuning based on both the specific steering objective and target model, necessitating empirical calibration to achieve the desired intervention strength.
>         3. **Monitoring:** When applying steering interventions, practitioners should monitor both the immediate output and latent space representations to ensure meaningful transformations while maintaining output coherence.
>
> ### References
> 1. Shi, Xiaoming, et al. "Time-MoE: Billion-Scale Time Series Foundation Models with Mixture of Experts." arXiv preprint arXiv:2409.16040 (2024).

---

> > ### Author Response · Authors · 2024-11-22
> > **Changes in response to your feedback (2/2)**
> >
> > 3. **Experiments on tasks beyond forecasting:**
> >     1. **Steering experiments:** In addition to forecasting ($\texttt{Chronos-Large}$) and reconstruction ($\texttt{MOMENT-Large}$), we conducted steering experiments on a classification task. Specifically, we used $\texttt{MOMENT-Large}$ to classify abnormal versus normal heartbeats on the real-world ECG5000 dataset. As shown in Appendix E.1 and illustrated in Figure 17, we demonstrate how steering can effectively intervene to change classifications (e.g., steering predictions to classify abnormal heartbeats as normal and vice versa).
> >     2. **Pruning experiments:** Beyond forecasting and reconstruction, we conducted pruning experiments on classification and anomaly detection tasks. Results were added to Appendix E.2, see Figure 18, and Tables 7-10 specifically.
> > 4. **Computational Overhead of Computing and Applying Steering Matrices:** Steering matrices are pre-computed and only applied during inference. The wall clock times reported in Appendix E.1 were measured for real-world ECG steering experiments conducted on a standard computing cluster with AMD EPYC 7502 CPUs. These timings include not only the application of steering matrices but also the setup overhead, such as initializing the model and forward hooks, in addition to the inference time.
> >     1. **Computing steering matrices:** Given a dataset with N examples, calculating the mean or median activations for a specific layer takes $O(N)$ or $O(N \log N)$ time, respectively. For the real-world ECG steering experiments, computing the steering matrix took $7.01 s \pm 25.2$ ms per loop (mean $\pm$ std. dev. of 7 runs, 1 loop each).
> >     2. **Applying steering matrices:** Applying steering matrices during inference is an $O(1)$ operation. For MOMENT-Large, the average inference time without intervention was $4.34 s \pm 51.2$ ms per loop, while interventions increased the time marginally to $4.39 s \pm 42$ ms per loop (mean $\pm$ std. dev. of 7 runs, 1 loop each).
> >     3. These results indicate that steering adds minimal computational overhead while enabling effective control over model predictions.
> >
> > Once again, we sincerely thank you for your time and thoughtful feedback, which resulted in us conducting more extensive experimentation. We kindly ask you to consider revising your score in light of these additions and improvements.

---

> > > ### Comment · Reviewer_QT6A · 2024-11-26
> > >
> > > Thanks for the authors reply. They have answered most of my questions. I will maintain my score.

---

> > > > ### Author Response · Authors · 2024-11-26
> > > > **Thanks for your feedback!**
> > > >
> > > > Dear Reviewer QT6A,
> > > >
> > > > Thank you for your thoughtful questions and for engaging with us. We are glad to hear that our response has addressed most of your concerns. If there are any remaining points or questions that require further clarification or elaboration, we would be more than happy to address them. Once again, we greatly appreciate your constructive input, which has helped improve the quality of our paper.

---

### Official Review · Reviewer_kpxg · 2024-11-08

**Soundness:** 2
**Presentation:** 2
**Contribution:** 3
**Rating:** 6
**Confidence:** 3

**Summary:**

This paper aims to understand the internal representations of time series foundation models (TSFM). By analyzing the self-similarity of the model layers, this paper reveals block-like redundancy in the structure that can be used for model pruning. This representational analysis also shows that the conceptual steering  potentially enables more controllable analysis for TSFM.

**Strengths:**

1.	The main aim of this paper is to address the gap in understanding the underlying mechanisms and learned representations of TSFMs, which remain largely unexplored. The representational analysis examined the representational similarity based on Centered Kernel Alignment (CKA), identified concepts, and localized them to specific hidden states. The representation redundancy found in this analysis was used to prune the model, which reduces the model size as well as improves the inference speed.
2.	The findings in this paper help us understand the TSFMs, and the knowledge can potentially be used to improve their capabilities. For example,  the predictions can be steered along the conceptual directions using synthetic time series.

**Weaknesses:**

1.	The analysis is conducted using many existing techniques, which are not specifically designed for time series models. There may be room for improvement by introducing novel techniques specifically designed for TSFMs.
2.	Section 3.2 is a huge subsection that contains many paragraphs. These paragraphs can be further grouped into subsubsections for better viewing.

**Questions:**

1. Applying concept steering interventions across all tokens is necessary to achieve the intended steered concept output compared to applying concept steering interventions to a single token. What is the main purpose of concept steering? Is it controllable to achieve some target objectives?

---

> ### Author Response · Authors · 2024-11-20
> **Changes in response to your feedback!**
>
> Dear Reviewer kpxg,
>
> Thank you for taking the time to review our paper and provide thoughtful feedback. We greatly appreciate your recognition of our contributions, particularly your observation that our findings “help us understand the TSFMs, and the knowledge can potentially be used to improve their capabilities.”
>
> You have highlighted two key areas for improvement: (1) the development of analysis techniques tailored to time series models, and (2) enhancing the readability of Section 3.2. We have carefully considered your suggestions and made revisions to address these concerns. Below, we detail our responses and summarize the changes implemented in the paper. All revisions are highlighted in blue ink for ease of review.
>
> 1. **Based on your suggestions, we have re-organized Section 3.2 by using `\subsubsections` to improve readability.**
> 2. **Regarding the design of time series specific techniques:** We appreciate your feedback and would like to clarify the following points:
>     1. The **primary goal of our study was to uncover the inner workings of time series foundation models** (TSFMs) and leverage these insights to enhance their capabilities and trustworthiness. We are pleased that you acknowledged the importance of this effort, noting that the “findings in this paper help us understand [the] TSFMs, and the knowledge can potentially be used to improve their capabilities.”
>     2. **Most TSFMs are based on the Transformer architecture:** Given that most TSFMs, except Tiny Time Mixers, are built on the transformer architecture, we focused on established analysis techniques for transformer architectures. Our work is the first to adapt these existing techniques to transformer-based TSFMs.
>     3. **Our findings with standard techniques are also valuable:** The success of using general transformer probing techniques in studying TSFMs is itself a key finding, as it highlights that: (1) the fundamental mechanisms of representation learning in transformers are largely consistent across modalities, (2) TSFMs learn structured, interpretable concepts, akin to their language and vision counterparts, and (3) certain techniques for studying transformer architectures are effective across modalities.
>     4.  **Our experiments are specific to time series:** Nonetheless, our study remains firmly rooted in time-series-specific phenomena and concepts: we employed time series models ($\texttt{MOMENT, Chronos, MOIRAI}$), evaluated them on time series tasks (forecasting, classification, anomaly detection, and imputation), and explored concepts central to time series data, such as periodicity, trend, and amplitude.
>     5. That said, we agree that **developing modality-specific probing techniques is a promising research direction.** We look forward to exploring how such techniques can further enrich the understanding and utility of TSFMs.

---

> > ### Author Response · Authors · 2024-11-20
> > **Answer to your questions!**
> >
> > 1. **What is the main purpose of concept steering?** Below is a summary of how concept steering can be useful in practical applications, as discussed in Section 5:
> >     1. **Introducing Inductive Biases:** Concept steering can embed inductive biases into model predictions, which is particularly beneficial for pre-trained models that may lack exposure to certain concepts due to limited or restricted training data. This approach is especially useful in domains such as healthcare. For example, domain knowledge about the excitatory or inhibitory effects of treatments can guide predictions about whether a patient’s vital signs should increase or decrease in response to specific interventions.
> >     2. **Correcting Prediction Errors:** Concept steering can address prediction errors arising from out-of-distribution inference, exogenous factors, or incomplete prior knowledge. For instance, in financial forecasting, the price of a stock might rise after a company exceeds market expectations in its earnings report. If a TSFM is unaware of this external information, domain experts can apply trend steering to bias forecasts with upward trends, ensuring more accurate predictions.
> >     3. **Data Augmentation:** Concept steering can also function as a data augmentation technique, allowing modifications to time series data in meaningful ways. Prior research (e.g., Ansari et al., 2024) has demonstrated that data augmentation improves TSFM generalization by enhancing robustness and exposing models to diverse patterns. For example, concept steering can simulate abnormal heartbeats in time series data for a healthy patient, expanding the range of patterns the model learns.
> >
> > 2. **Is it controllable to achieve some target objectives?** Thank you for your excellent question. Yes, concept steering can be controlled to meet target objectives, to some extent. For example, the parameter $\alpha$ can be used to control the degree of trend or seasonality added to a time series prediction, as demonstrated in Figure 9. Additionally, we show how multiple concepts can be composed and how varying levels of trend and seasonality can be applied to time series forecasts and reconstructions, all within the same figure. That said, achieving finer-grained control over steering remains an exciting and important avenue for future research. For instance, it would be interesting to explore how to control the slope of trends or adjust the seasonality of forecasts from weekly to monthly patterns.
> >
> > We made the discussion clearer by adding these points.
> >
> > Please feel free to reach out if you have any further questions. We would also greatly appreciate it if you could consider revising your score based on the changes to the paper. Thank you again for your time and careful attention to our work.

---

> > > ### Comment · Reviewer_kpxg · 2024-11-26
> > >
> > > Thank you for your responses. Since most of my concerns have been addressed, I have increased my score to 6.

---

> > > > ### Author Response · Authors · 2024-11-26
> > > > **Thank you so much for your feedback!**
> > > >
> > > > Dear Reviewer kpxg,
> > > >
> > > > Thank you for your thoughtful feedback and for raising the score of our manuscript. We are glad to hear that our response has addressed most of your concerns. If there are any remaining points or questions that require further clarification or elaboration, we would be more than happy to address them. Once again, we greatly appreciate your constructive input, which has helped improve the quality of our paper.

---

### Author Response · Authors · 2024-11-22
**Global Response**

Dear Reviewers and Area Chair,

We sincerely appreciate the time and effort you have dedicated to reviewing our paper. Below, we summarize the strengths of our work as identified by the reviewers:
1. **Meaningful Contributions:** Reviewer MRNR remarked that our work is “very meaningful, being at a time when TSFMs are emerging, it is crucial to explore more efficient, interpretable, and controllable models.” Our work not only addresses timely challenges in an emerging area of research but also demonstrates practical utility, as noted by reviewers kpxg and QT6A, who remarked that our methods “effectively reduce model size and improve inference speed while maintaining accuracy.” We are encouraged that all reviewers (kpxg, QT6A, and MRNR) consider our work a "good" contribution, as reflected in their ratings.
2. **Novel Methods:** Reviewer kpxg recognized as a strength that our work “addresses the gap in understanding the underlying mechanisms and learned representations of TSFMs, which remain largely unexplored”. Similarly, reviewer GT6A recognizes the novelty of our work and mentions that “By comparing representational similarity between layers of the model, this work provides a new method for analyzing TSFMs” and “It introduces the ability to steer model predictions along conceptual directions, influencing outputs in targeted ways.”:

3. **Generalizable and Reproducible Findings:** Reviewer QT6A appreciated that our work “is not limited to a single model but spans multiple TSFMs, enhancing the generality of the findings. The use of open-source models and data facilitates community reproduction and further research.” Additionally, reviewer kpxg noted that our findings “help us understand TSFMs and can potentially be used to improve their capabilities,” while MRNR highlighted their utility in providing “guidance for future TSFM architecture design.”

While these strengths are encouraging, your thoughtful questions and suggestions helped us identify key areas for improvement. We summarize the revisions and responses below:

1. **Improve readability:** Reviewers kpxg and MRNR highlighted the need for better organization, particularly in Section 3.2, and a more comprehensive introduction. In response, we made substantial revisions to improve the clarity, structure, and flow of the paper.
2. **Computational Overhead of Steering:** Reviewer QT6A expressed concern over the computational overhead of computing and applying steering matrices. Our analysis, detailed in the response to QT6A, demonstrates that steering introduces minimal computational overhead while enabling effective control of model predictions.
3. **Experiments on tasks beyond forecasting:** Reviewer QT6A also noted the need for broader experimentation. In the revised manuscript, we added pruning experiments on classification and anomaly detection, in addition to imputation/reconstruction, and forecasting. We also added steering experiments for classification on a real-world ECG dataset in addition to imputation/reconstruction, and forecasting.

---

### Author Response · Authors · 2024-11-24
**Author Reviewer Discussion**

Dear Reviewers and Area Chair,

We hope you are doing well!

We really appreciate your time and feedback! As the author-reviewer discussion period draws to end, we were hoping to engage with you to address any remaining questions or concerns.

We have made several changes to the paper in response to your feedback as summarised in the global response.

Thanks for helping us improve our paper and looking forward to hearing from you.

Authors

---

### Meta-Review · Area_Chair_Mpde · 2024-12-21

**Metareview:**

This work examines the internal representations of Time Series Foundation Models (TSFMs), identifying block-like redundancy within these representations. By exploiting this redundancy, the authors propose an informed pruning strategy that enhances inference speed and efficiency without sacrificing accuracy. Additionally, the study investigates learned concepts such as periodicity and trends, demonstrating the potential of latent space steering to manipulate these concepts and influence model behavior. This approach enables the introduction of new features to time series signals, offering a novel way to guide model predictions in specific conceptual directions.

While the method effectively reduces model size, improves inference speed, and maintains accuracy, a significant concern is that the paper lacks a cohesive logical structure and reads more like a technical report. Furthermore, it fails to offer any substantial new insights to the research community.

**Additional Comments On Reviewer Discussion:**

The reviewers raised concerns about the paper's poor presentation, disorganized structure, questionable parameter choices, computational overhead, and unclear motivation. While the authors have made efforts to improve the paper's readability, these changes were not sufficient to address the core issues. The reviewers did not express significant support for the work, and the overall quality of the paper remains below the threshold for acceptance.

---

### Decision · Program_Chairs · 2025-01-22

Reject